# Allosteric rescue of catalytically impaired ATP phosphoribosyltransferase variants links protein dynamics to active-site electrostatic preorganisation

Gemma Fisher[1], Marina Corbella [2], Magnus S. Alphey[1], John Nicholson[1], Benjamin J. Read[1], Shina C. L. Kamerlin [2,3] ✉ & Rafael G. da Silva [1] ✉

ATP phosphoribosyltransferase catalyses the first step of histidine biosynthesis and is controlled via a complex allosteric mechanism where the regulatory protein HisZ enhances catalysis by the catalytic protein $HisG_S$ while mediating allosteric inhibition by histidine. Activation by HisZ was proposed to position $HisG_S$ Arg56 to stabilise departure of the pyrophosphate leaving group. Here we report active-site mutants of $HisG_S$ with impaired reaction chemistry which can be allosterically restored by HisZ despite the $HisZ:HisG_S$ interface lying ~20 Å away from the active site. MD simulations indicate HisZ binding constrains the dynamics of $HisG_S$ to favour a preorganised active site where both Arg56 and Arg32 are poised to stabilise leaving-group departure in $WT-HisG_S$. In the $Arg56Ala-HisG_S$ mutant, HisZ modulates Arg32 dynamics so that it can partially compensate for the absence of Arg56. These results illustrate how remote protein-protein interactions translate into catalytic resilience by restoring damaged electrostatic preorganisation at the active site.

Robust reaction rate enhancement and allosteric regulation are hallmarks of enzyme catalysis, and both aspects may be at least in part underpinned by protein conformational flexibility[1–3]. The catalytic prowess of enzymes can be significantly ascribed to substrate binding to an electrostatically preorganised active site, which minimises the reorganisation energy required for optimum stabilisation of the charge redistribution as the reaction progresses from the reactant state to the transition state[4]. Yet several lines of evidence also suggest a contribution from protein dynamics[5–8], from nonstatistical, femtosecond-timescale vibrations coupled directly to transition-state barrier crossing[6,8], to slower, thermally equilibrated motions reshaping the enzyme conformational ensemble towards populations where active-site preorganisation is optimised[5]. Nonetheless, this topic is still controversial possibly due to the inherent flexibility of proteins which makes it difficult to isolate motions that may have evolved to facilitate reaction[7,9,10].

Allosteric modulation of enzymes, i.e. the alteration of reaction rate and/or substrate affinity upon ligand binding to, mutation of, or post-translational modification at a site remote from the active site, is a fundamental regulatory mechanism of biochemical reactions[11,12]. It finds applications in drug discovery to facilitate drug-target selectivity as allosteric sites tend to be less conserved than active sites across homologous proteins[13,14], and in enzyme engineering and synthetic biology, where allosteric control may need to be introduced or, more often, eliminated[15,16]. While enzymes subject to allosteric regulation by ligand binding can be broadly classified as $K$-type, those where substrate affinity is altered, and $V$-type, those where the steady-state catalytic rate constant ($k_{cat}$) is altered, the specific kinetic steps affected can vary depending on the enzyme[13,17–19]. For instance, in *Mycobacterium tuberculosis* α-isopropylmalate synthase, the rate-limiting step changes from product release to chemistry upon allosteric inhibition

[1]School of Biology, Biomedical Sciences Research Complex, University of St Andrews, St Andrews KY16 9ST, UK. [2]Science for Life Laboratory, Department of Chemistry – BMC, Uppsala University, S-751 23 Uppsala, Sweden. [3]School of Chemistry and Biochemistry, Georgia Institute of Technology, 901 Atlantic Drive NW, Atlanta, GA 30332, USA. ✉e-mail: lynn.kamerlin@kemi.uu.se; rgds@st-andrews.ac.uk

by leucine[17]. The role of protein dynamics in allostery has been much less controversial when discussed in terms of conformational changes to promote physical events such as substrate binding and product release, or the interconversion rate among conformations[11,20]. However, in systems where allosteric regulation affects the rate of the chemical step itself[17–19], the intersection at which local and remote protein motions, active-site electrostatic preorganisation, and ultimately catalysis meet remains challenging to pinpoint, despite recent advances toward this goal with Kemp eliminase[5,21].

ATP phosphoribosyltransferase (ATPPRT) (EC 2.4.2.17) catalyses the $Mg^{2+}$-dependent formation of $N^1$-(5-phospho-β-D-ribosyl)-ATP (PRATP) and inorganic pyrophosphate ($PP_i$) from ATP and 5-phospho-α-D-ribose 1-pyrophosphate (PRPP) (Fig. 1a), the first and flux-controlling step of histidine biosynthesis, and is allosterically inhibited by histidine in a negative feedback control loop[22,23]. ATPPRT is the focus of synthetic biology efforts to enable the production of histidine in bacteria[16,24] and a promising drug target against some pathogenic bacteria[25–27]. Short-form ATPPRTs form an intricate allosteric system comprising catalytic ($HisG_S$) and regulatory (HisZ) subunits assembled as a hetero-octamer with a tetrameric core of HisZ sandwiched by two dimers of $HisG_S$[28–33]. $HisG_S$ on its own is catalytically active and insensitive to inhibition by histidine[28,34]. Binding of HisZ, which has no catalytic power of its own, to form the ATPPRT holoenzyme allosterically activates catalysis by $HisG_S$[26,28,34]. However, HisZ also harbours the pocket where histidine binds and allosterically inhibits ATPPRT. Thus, the regulatory protein plays a dual role, as allosteric activator of catalysis in the absence of histidine and mediator of allosteric inhibition in the presence of histidine[19,31,33].

In *Psychrobacter arcticus* ATPPRT (*Pa*ATPPRT), structural and functional data indicate allosteric activation of catalysis triggered by HisZ (*Pa*HisZ) binding specifically perturbs the chemical step of the reaction taking place at the $HisG_S$ (*Pa*HisG_S) active site located ~23 Å from the nearest *Pa*HisG_S:*Pa*HisZ interface[18,29]. First, crystal structures of the Michaelis complexes of the activated, hetero-octameric holoenzyme (henceforth referred to as *Pa*ATPPRT) and of the non-activated, dimeric enzyme (henceforth referred to as *Pa*HisG_S) showed Arg56 of one of the catalytic subunits reaching across the dimer interface to form a salt-bridge with the pyrophosphate moiety of PRPP in the active site of the other subunit (Fig. 1b) in the *Pa*ATPPRT structure but not in the *Pa*HisG_S structure. Therefore, allosteric activation was proposed to lead to more efficient leaving group departure at the transition state by stabilisation of the negative charge build-up on the pyrophosphate upon nucleophilic attack of ATP N1 on PRPP C1[29]. Second, no burst in product formation was observed for the reaction catalysed by *Pa*HisG_S, and the multiple-turnover, pre-steady-state rate constant was in agreement with $k_{cat}$, suggesting chemistry is rate limiting in the nonactivated enzyme reaction. In contrast, a burst was observed in the reaction catalysed by *Pa*ATPPRT, producing a rate-constant ($k_{burst}$) much higher than $k_{cat}$, supporting a mechanism where allosteric activation speeds up the chemical step, making product release rate limiting[18]. Finally, replacement of $Mg^{2+}$ by $Mn^{2+}$, which more efficiently offsets the negative charge at the transition state, led to an ~3-fold enhancement of *Pa*HisG_S $k_{cat}$, as would be predicted, qualitatively, for a rate-limiting chemical step, but had no effect on *Pa*ATPPRT $k_{cat}$, where chemistry was already much faster than subsequent steps[18].

As expected, an R56A-*Pa*HisG_S mutant had a reduced reaction rate in the nonactivated enzyme as measured at a fixed concentration of substrates, since R56 was posited to be important for leaving group departure in *Pa*HisG_S as well, only less efficiently. Intriguingly, upon *Pa*HisZ binding to R56A-*Pa*HisG_S, part of the activity was recovered[29]. Here we employed site-directed mutagenesis, differential scanning fluorimetry (DSF), enzyme kinetics, [31]P-NMR spectroscopy, protein crystallography, and molecular dynamics (MD) simulations to dissect this phenomenon, reveal that other single- and double-mutations at the *Pa*HisG_S active site display similar behaviour, and demonstrate

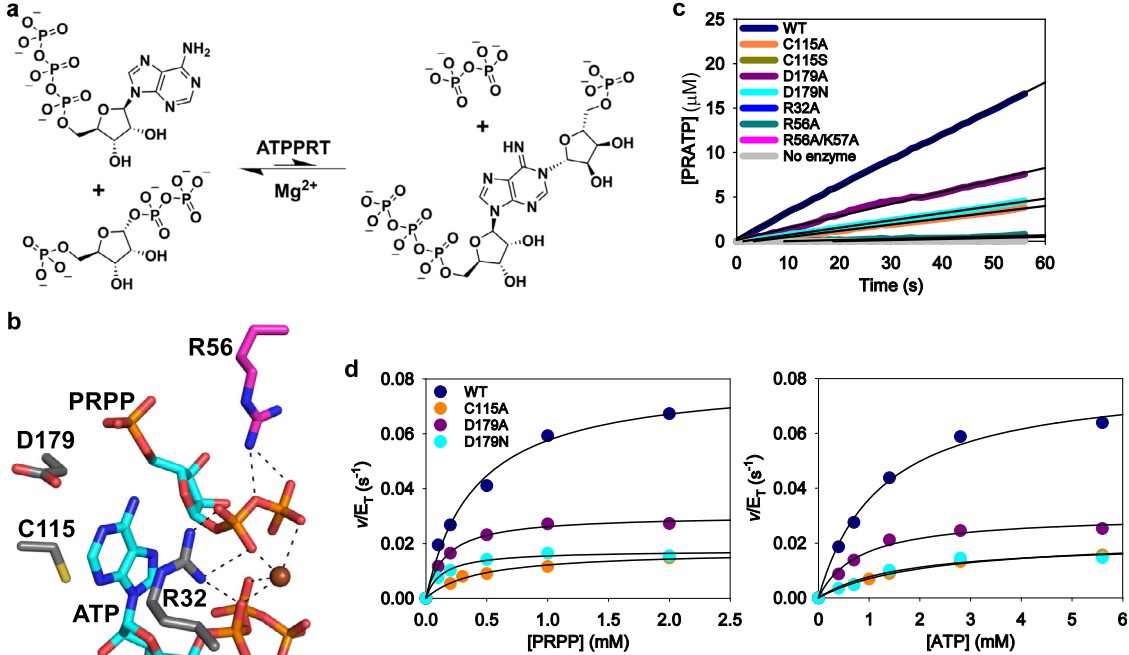

**Fig. 1 | The effect of active-site mutations on *Pa*HisG_S. a** The reversible, $Mg^{2+}$-dependent reaction catalysed by ATPPRT. **b** Stick model of the active site of *Pa*ATPPRT (PDB ID 6FU2)[29]. Oxygen is shown in red, nitrogen in blue, phosphorus in orange, sulphur in yellow, and carbon in either cyan (substrates), grey (in one of the *Pa*HisG_S subunits) or magenta (in the other *Pa*HisG_S subunit). Polar interactions are depicted as dashed lines, and the $Mg^{2+}$ as sphere. **c** Representative traces of PRATP formation time course catalysed by 5 μM *Pa*HisG_S variants. Black lines are linear regressions of the data. Source data are provided as a Source Data file. **d** Substrate saturation curves for WT and mutant *Pa*HisG_S. Data are the mean of two independent measurements. Lines are best fit of the data to Eq. (1). Source data are provided as a Source Data file.

how modulation of $Pa$HisG$_S$ dynamics by $Pa$HisZ propagates to the active site to affect the chemical step.

## Results

### C115-, D179A-, and D179N-$Pa$HisG$_S$ are catalytically active

To assess the importance of active-site residues in catalysis, we introduced single and double mutations into the $Pa$HisG$_S$ amino acid sequence by site-directed mutagenesis of the $Pa$HisG$_S$-coding sequence and expressed and purified the mutant proteins (Supplementary Fig. 1). Electrospray ionisation/time-of-flight-mass spectrometry (ESI/TOF-MS) confirmed the molecular masses of WT-, C115A-, C115S-, D179A-, R32A-, R56A-, and R56A/K57A-$Pa$HisG$_S$ variants were in agreement with the predicted values (Supplementary Fig. 2). The introduction of the D179N mutation was confirmed by MS/MS analysis of tryptic fragments (Supplementary Fig. 2). In an initial activity screen, reactions were monitored for just under 1 min by the continuous and direct UV/VIS absorbance-based assay for ATPPRT activity[18,35] at fixed PRPP and ATP concentrations sufficient to saturate WT-$Pa$HisG$_S$. PRATP formation was readily detected with C115A-, D179A-, D179N-, and WT-$Pa$HisG$_S$ (Fig. 1c), and linear regression of the data yielded apparent rate constants shown in Supplementary Table 1.

Substrate saturation curves for WT-, C115A-, D179A-, and D179N-$Pa$HisG$_S$ obeyed Michaelis-Menten kinetics (Fig. 1d), and data fit to Eq. (1) produced the apparent steady-state kinetic parameters in Supplementary Table 2. The Michaelis constant for ATP ($K_{ATP}$) increased less than 2-fold for C115A-$Pa$HisG$_S$, and the Michaelis constant for PRPP ($K_{PRPP}$) actually decreased between 2- and 3-fold for D179A- and D179N-$Pa$HisG$_S$, suggesting C115 and D179 make negligible contributions to substrate binding. The $k_{cat}$ values for the mutants were reduced only ~4-fold in comparison with the WT-$Pa$HisG$_S$, pointing to these residues' modest importance in catalysis. Over the course of the ATPPRT-catalysed reaction, the 6-NH$_2$ group must donate a proton to yield the 6-NH group of PRATP, and in the case of $Pa$ATPPRT, this proton abstraction happens on the enzyme[18]. Based on their respective positions in the active site (Fig. 1b)[29], both C115 and D179 were candidates to act as general base for this proton abstraction, but the small catalytic effect of their replacements for residues that cannot participate in acid-base catalysis does not support such role, leaving the identity of the general base still elusive.

### C115S-, R32A-, R56A-, and R56A/K57A-$Pa$HisG$_S$ are catalytically compromised

No PRATP formation could be detected above the background noise of the assay (no-enzyme control) during the initial activity screen when either C115S-, R32A-, R56A-, or R56A/K57A-$Pa$HisG$_S$ was used as catalyst (Fig. 1c), indicating these mutants have impaired catalytic activity. DSF-based thermal denaturation assays showed these mutants display similar thermal unfolding profiles to the WT protein (Fig. 2a), demonstrating the mutations do not thermally destabilise the tertiary structure of the protein, and data fit to Eq. (2) yielded melting temperatures ($T_m$) shown in Supplementary Table 3. Moreover, as described previously[18] and repeated here for WT-$Pa$HisG$_S$, the presence of PRPP increased the $T_m$ of the mutants (Fig. 2a; Supplementary Table 3), indicating the catalytically impaired $Pa$HisG$_S$ variants can bind PRPP, in agreement with the ordered kinetic mechanism proposed for this enzyme[18,29]. Analytical size-exclusion chromatography produced similar elution profiles for WT-, C115S-, R32A-, R56A-, and R56A/K57A-$Pa$HisG$_S$ (Supplementary Fig. 3), which includes the expected dimer[29] and a higher oligomeric state. An activity screen with longer reaction times and twice as much enzyme as in the initial screen, to allow more product to accumulate, demonstrated the catalytic ability of C115S-, R32A-, R56A-, and R56A/K57A-$Pa$HisG$_S$ is significantly diminished but not abolished (Fig. 2b).

Even though C115 is only modestly important for catalysis, its replacement by serine led to a 117-fold reduction in activity (Table 1), perhaps due to the introduction of a detrimental interaction. The activities of R32- and R56A-$Pa$HisG$_S$ decreased 25- and 42-fold, respectively, in comparison with the WT-$Pa$HisG$_S$ (Table 1). This demonstrates the importance of these residues in $Pa$HisG$_S$ catalysis, possibly because R56 and R32 may contribute to leaving group departure at the transition state[18,29]. K57 is adjacent to R56 in the $Pa$HisG$_S$ primary sequence, but in all $Pa$HisG$_S$ and $Pa$ATPPRT crystal structures, it points away from the active site[19,28,29]. We hypothesised that in the absence of the R56 guanidinium group, the K57 ε-NH$_3^+$ group could move towards the active site and assist in leaving group departure. However, the R56A/K57A-$Pa$HisG$_S$ double mutant displayed a 254-fold decrease in activity (Table 1), which is only ~6-fold more catalytically impaired than the R56A-$Pa$HisG$_S$, indicating just a modest catalytic importance for K57. R32, R56, K57, and C115 are highly conserved in HisG$_S$ across species, and D179 is also conserved but sometimes replaced with a glutamate residue[29]. Nevertheless, out of these five residues in $Pa$HisG$_S$, only the arginine residues seem to be significantly important for catalysis.

### $Pa$HisZ allosterically rescues C115S-, R32A-, R56A-, and R56A/K57A-$Pa$HisG$_S$ catalysis

To assess the extent to which these mutations were also detrimental to $Pa$ATPPRT catalysis, and how much of the R56A-$Pa$HisG$_S$ activity could be recovered in the presence of $Pa$HisZ[29], the effect of $Pa$HisZ on the reaction catalysed by the impaired $Pa$HisG$_S$ mutants was determined. The regulatory protein surprisingly led to activation of all catalytically impaired $Pa$HisG$_S$ mutants (Fig. 2c), and data fit to Eq. (3) resulted in the apparent equilibrium dissociation constants ($K_D$) for $Pa$HisZ displayed in Table 2. No activity was detected when the $Pa$HisG$_S$ mutant-catalysed reactions were carried out in the presence of bovine serum albumin (BSA) (Supplementary Fig. 4), ruling out that allosteric rescue was due to nonspecific protein binding. To gather orthogonal evidence for the allosteric rescue, the reaction catalysed by each $Pa$HisG$_S$ mutant was analysed by $^{31}$P-NMR spectroscopy in the presence and absence of $Pa$HisZ (Supplementary Fig. 5) under conditions where product can be detected with WT-$Pa$HisG$_S$[18]. The characteristic chemical shift at ~3.30 ppm (Fig. 2d) previously assigned to the phosphorus in the $N^1$-5-phospho-β-D-ribose moiety of PRATP[28,29], was only detected here when $Pa$HisZ was present in the reaction, confirming the rescue of the catalytically compromised mutants by the regulatory protein.

Substrate saturation curves for WT-, C115S-, R32A-, R56A-, and R56A/K57A-$Pa$ATPPRT obeyed Michaelis-Menten kinetics (Fig. 2e), and data fit to Eq. (1) produced the apparent steady-state kinetic parameters shown in Table 2, with the concentrations of each $Pa$ATPPRT variant calculated from the $K_D$ for $Pa$HisZ using Eq. (4). $Pa$HisZ allosterically restored most of the catalytic activity of the impaired $Pa$HisG$_S$ mutants. In comparison with the WT-$Pa$ATPPRT, $K_{PRPP}$ was unaltered by the mutations, and $K_{ATP}$ increased by a maximum of 4-fold. The $k_{cat}$ decreased by less than 2-fold for C115S-$Pa$ATPPRT as compared with WT-$Pa$ATPPRT (it is possible $Pa$HisZ binding allosterically disrupts a putative catalytically detrimental interaction involving S115), and by less than 4-fold and 6-fold for R32A- and R56A-$Pa$ATPPRT. Only R56A/K57A-$Pa$ATPPRT $k_{cat}$ was reduced by more than one order of magnitude (~14-fold) in comparison with WT-$Pa$ATPPRT, which is still a small effect in comparison with the 254-fold catalytic impairment of R56A/K57A-$Pa$HisG$_S$.

Due to the long TEVP-cleavage time and low yield of the $Pa$HisZ recovered, along with the extensive use of the regulatory protein in this work, a His-tagged $Pa$HisZ was purified and employed from this point onwards. The apparent steady-state kinetic parameters (Supplementary Fig. 6) are very similar whether nor not the His-tagged $Pa$HisZ was used. Histidine binds to $Pa$HisZ and allosterically inhibits $Pa$ATPPRT catalysis, and the suppression of the burst in product formation in the presence of histidine suggests that allosteric inhibition

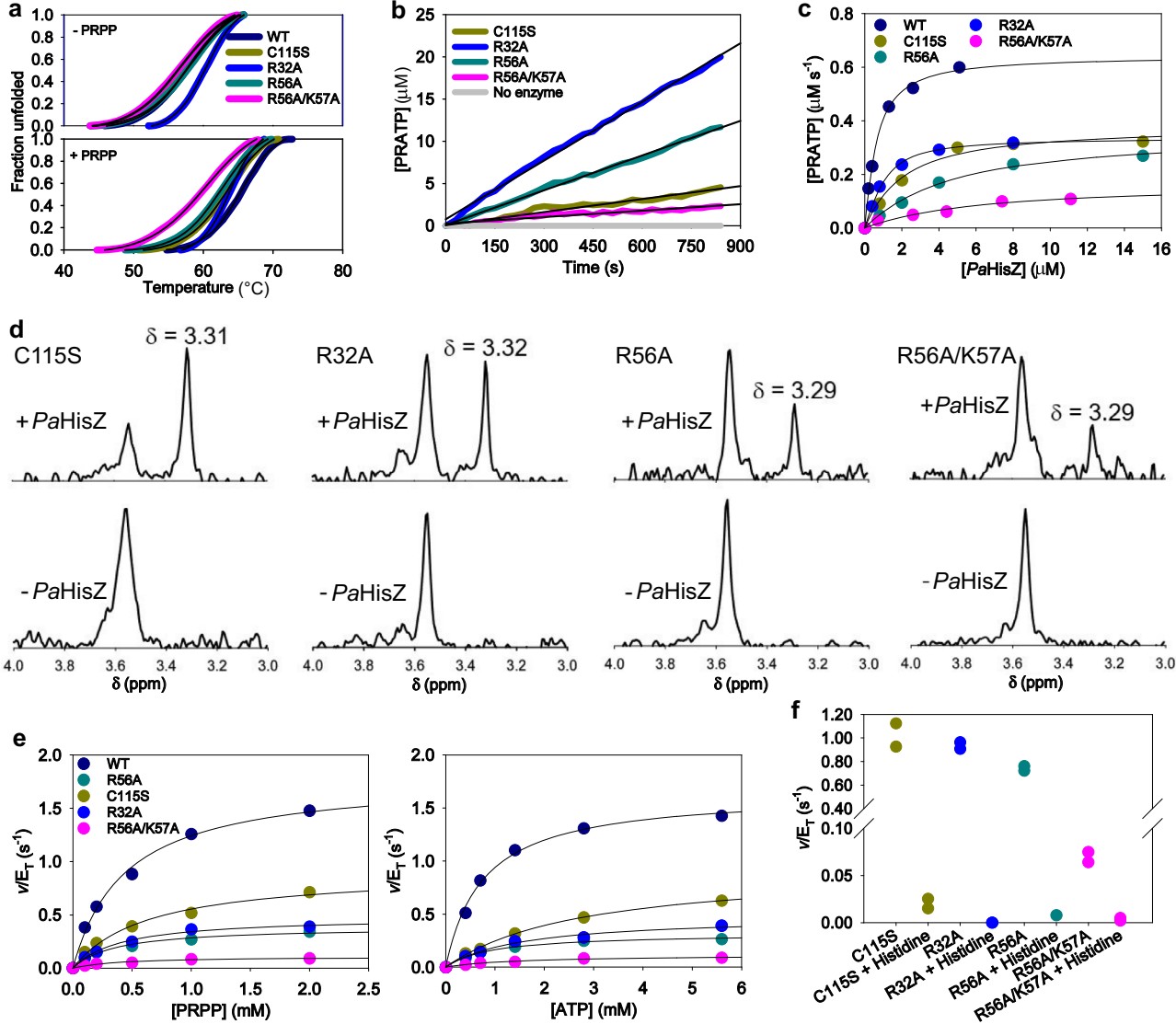

**Fig. 2 | Allosteric rescue of catalytically impaired *Pa*HisG$_S$ mutants by *Pa*HisZ.** **a** DSF-based thermal denaturation of C115S-, R32A-, R56A-, or R56A/K57A-*Pa*HisG$_S$. Traces are averages of three independent measurements. Lines of best fit to Eq. (2) are in black. Source data are provided as a Source Data file. **b** PRATP formation time course catalysed by 10 µM *Pa*HisG$_S$ variants. Traces are averages of two independent measurements. Black lines are linear regressions of the data. Source data are provided as a Source Data file. **c** Dependence of rate of reaction catalysed by *Pa*HisG$_S$ variants on the concentration of *Pa*HisZ. Data are the mean of two independent measurements. Lines are best fit of the data to Eq. (3). Source data are provided as a Source Data file. **d** Close-ups of the 4.0–3.0 ppm region of the ³¹P-NMR spectra of the reaction catalysed by *Pa*HisG$_S$ mutants in the presence and absence of *Pa*HisZ. The peak at ~3.30 ppm corresponds to the phosphorus in the $N^1$-5-phospho-β-D-ribose moiety of PRATP[28, 29]. Source data are provided as a Source Data file. **e** Substrate saturation curves for WT and mutant *Pa*ATPPRT. Data are the mean of two independent measurements. Lines are best fit of the data to Eq. (1). Source data are provided as a Source Data file. **f** Effect of 1 mM histidine on mutant *Pa*ATPPRT-catalysed reaction. Two independent measurements were carried out, and all data points are shown. Source data are provided as a Source Data file.

directly affects the chemical step of the reaction[19]. Histidine also inhibits the reaction catalysed by the rescued *Pa*ATPPRT mutants (Fig. 2f; Supplementary Fig. 7), indicating the allosteric pathway responsible for inhibition is intact in these mutants.

**Table 1 | Apparent rate constants (mean ± fitting error) for *Pa*HisG$_S$ mutants from reactions monitored for 840 s**

| *Pa*HisG$_S$ | v/E$_T$ (s⁻¹) | Catalytic impairment[a] |
|---|---|---|
| C115S | 0.0005 ± 0.0001 | 117-fold |
| R32A | 0.0023 ± 0.0001 | 25-fold |
| R56A | 0.0014 ± 0.0001 | 42-fold |
| R56A/K57A | 0.00023 ± 0.00005 | 254-fold |

[a]Ratio of v/E$_T$ for WT-*Pa*HisG$_S$ (0.0586 ± 0.0002) to v/E$_T$ reported here.

## Allosteric activation of WT- and R56A-*Pa*HisG$_S$ by an orthologous HisZ

*Pa*HisZ and *Pa*HisG$_S$ share 43% and 69% sequence identity with their orthologues from the pathogenic bacterium *Acinetobacter baumannii*, *Ab*HisZ and *Ab*HisG$_S$, respectively, but *Pa*HisZ has been shown to be a potent allosteric inhibitor of *Ab*HisG$_S$[26]. We thus hypothesised that *Ab*HisZ could inhibit WT-*Pa*HisG$_S$. However, addition of *Ab*HisZ activated catalysis by WT-*Pa*HisG$_S$ (Fig. 3a), and data fit to Eq. (3) yielded an apparent $K_D$ for *Ab*HisZ of 9 ±1 µM. Moreover, *Ab*HisZ also rescued catalysis by R56A-*Pa*HisG$_S$ (Fig. 3a), but their interaction involved positive co-operativity as evidenced by the sigmoidal dependence of the reaction rate on the regulatory protein. The R56A-*Pa*HisG$_S$ mutant was chosen due to its significant catalytic impairment and the proposed role in catalysis for R56. The data were fit to Eq. (5), yielding a concentration of *Ab*HisZ at the inflection point ($K_{0.5}$) and a Hill

Table 2 | Apparent steady-state kinetic parameters (mean±fitting error) for *Pa*ATPPRT variants (all mutations are in *Pa*HisG$_S$)

| *Pa*ATPPRT | *Pa*HisZ $K_D$ (μM) | $k_{cat}$ (s$^{-1}$) | $K_{PRPP}$ (mM) | $K_{ATP}$ (mM) | Activation by *Pa*HisZ[a] |
|---|---|---|---|---|---|
| WT | 0.44 ± 0.05 | 1.72 ± 0.07 | 0.44 ± 0.05 | 0.76 ± 0.07 | 29-fold |
| C115S | 1.7 ± 0.4 | 0.93 ± 0.05 | 0.6 ± 0.1 | 2.8 ± 0.2 | 1,860-fold |
| R32A | 0.49 ± 0.06 | 0.48 ± 0.04 | 0.41 ± 0.06 | 1.5 ± 0.3 | 208-fold |
| R56A | 4.0 ± 0.6 | 0.35 ± 0.05 | 0.37 ± 0.04 | 0.9 ± 0.1 | 250-fold |
| R56A/K57A | 5 ± 2 | 0.12 ± 0.01 | 0.4 ± 0.1 | 1.5 ± 0.3 | 521-fold |

[a]Ratio of $k_{cat}$ reported here to $v/E_T$ from Supplementary Table 1 (WT) and Table 1 (mutants).

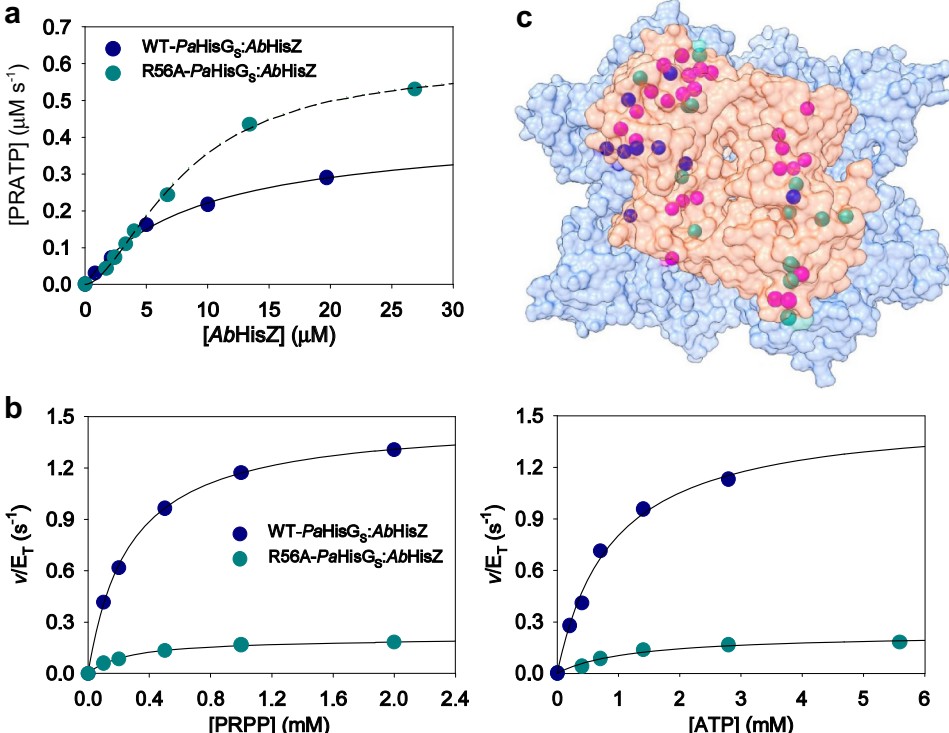

**Fig. 3 | Allosteric activation of PaHisG$_S$ variants by *Ab*HisZ. a** Dependence of rate of reaction catalysed by *Pa*HisG$_S$ variants on the concentration of *Ab*HisZ. Data are the mean of two independent measurements. Best fit of the data to Eq. (3) is shown as a solid line. Best fit of the data to Eq. (5) is shown as a dashed line. Source data are provided as a Source Data file. **b** Substrate saturation curves for WT-*Pa*HisG$_S$/*Ab*HisZ and R56A-*Pa*HisG$_S$/*Ab*HisZ. Data are the mean of two independent measurements. Lines are best fit of the data to Eq. (1). WT-*Pa*HisG$_S$/*Ab*HisZ concentration was calculated from the $K_D$ for *Ab*HisZ with Eq. (4), while R56A-*Pa*HisG$_S$/*Ab*HisZ concentration was assumed to be the same as R56A-*Pa*HisG$_S$ in the presence of 26 μM *Ab*HisZ. Source data are provided as a Source Data file. **c** The crystal structure of *Pa*ATPPRT[28] (PDB ID 5M8H) viewed from above the *Pa*HisG$_S$ dimer. *Pa*HisG$_S$ (orange) and *Pa*HisZ (blue) are in surface rendering. The Cα atoms of specific *Pa*HisZ residues at the interface with *Pa*HisG$_S$ are shown as spheres, with pink depicting identical residues to *Ab*HisZ, blue depicting residues with similar properties to those in *Ab*HisZ, and green, those not conserved in *Ab*HisZ.

coefficient (*h*) of 8.1 ± 0.4 μM and 1.68 ± 0.08, respectively. Nonetheless, this fit is intended only to highlight the sigmoidal behaviour of the data, since the experiment could not be carried out under pseudo-first-order conditions, i.e. [R56A-*Pa*HisG$_S$] ~ [*Ab*HisZ] in the experiment; thus, Eq. (5) does not hold.

Substrate saturation curves for WT-*Pa*HisG$_S$/*Ab*HisZ and R56A-*Pa*HisG$_S$/*Ab*HisZ hybrid complexes obeyed Michaelis-Menten kinetics (Fig. 3b), and data fit to Eq. (1) produced the following apparent $k_{cat}$, $K_{PRPP}$, and $K_{ATP}$: 1.49 ± 0.02 s$^{-1}$, 0.27 ± 0.01 mM, and 0.9 ± 0.1 mM for WT-*Pa*HisG$_S$/*Ab*HisZ; 0.22 ± 0.02 s$^{-1}$, 0.27 ± 0.02 mM, and 1.2 ± 0.3 mM for R56A-*Pa*HisG$_S$/*Ab*HisZ. These values are in good agreement with those for WT- and R56A-*Pa*ATPPRT (Table 2), indicating *Ab*HisZ recapitulates the catalytic activation of *Pa*HisG$_S$ to a similar extent the native *Pa*HisZ does. Furthermore, *Ab*HisZ is capable of efficiently relaying the histidine inhibition allosteric signal to WT- and R56A-*Pa*HisG$_S$ (Supplementary Fig. 8). We mapped the *Pa*HisZ residues at the interface between the *Pa*HisG$_S$ dimer and the *Pa*HisZ tetramer in the

*Pa*ATPPRT hetero-octamer[28] (Fig. 3c), and found that 25 of those are strictly conserved in *Ab*HisZ. It is possible that these residues have an important role in transmitting the allosteric signal for both activation and inhibition.

## Crystal structures of R56A-*Pa*HisG$_S$ and R56A-*Pa*ATPPRT

In an attempt to gain insight into the effect of the R56A mutation at the atomic level, we solved the crystal structures of both R56A-*Pa*HisG$_S$ and R56A-*Pa*ATPPRT bound to PRPP:ATP. Refinement statistics are summarised in Supplementary Table 4, and the electron-density maps for PRPP:ATP in both structures are shown in Supplementary Fig. 9. Even WT-*Pa*HisG$_S$ and WT-*Pa*ATPPRT are known to form the Michaelis complex *in crystallo*, likely due to a highly unfavourable on-enzyme equilibrium for the forward reaction[18,29]. The active site interactions are very similar between the two structures, except the electron density for Mg$^{2+}$ in R56A-*Pa*ATPPRT was not well defined, so the metal was not modelled in, and the hydrogen bond between E163 and the PRPP 3′-OH

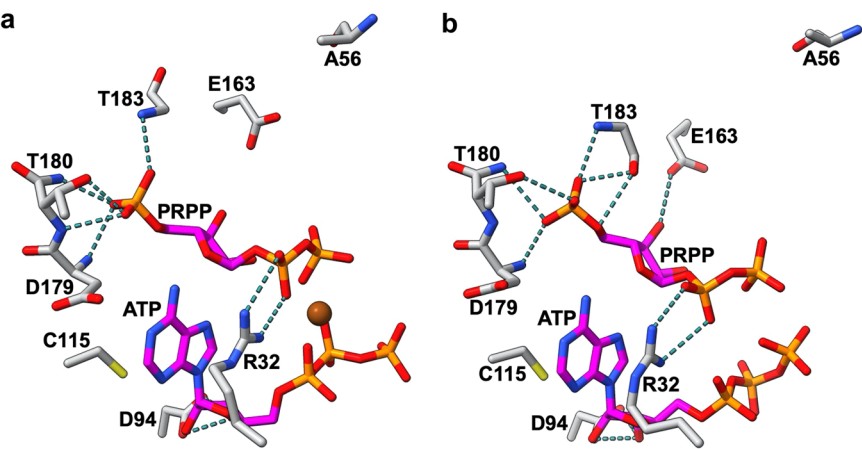

**Fig. 4 | Crystal structures of R56A-PaHisG$_S$ and R56A-PaATPPRT bound to PRPP:ATP. a** Stick model of the active site of R56A-PaHisGS. **b** Stick model of the active site of R56A-PaATPPRT. In both panels, oxygen is shown in red, nitrogen in blue, phosphorus in orange, sulphur in yellow, and carbon in either magenta (substrates) or grey (protein). The A56 side chain is contributed by the adjacent PaHisG$_S$ subunits. Polar interactions are depicted as dashed lines, and the Mg$^{2+}$ as a sphere.

is absent in R56A-PaHisG$_S$ (Fig. 4). The structures are also very similar to those of the respective WT enzymes[29]. Therefore, any structural or conformational differences that may lead to allosteric rescue of R56A-PaHisG$_Z$ catalysis are not captured in the static view of the crystal structure.

## PaHisZ binding alters PaHisG$_S$ dynamics

MD simulations were performed on the WT-PaHisG$_S$:PRPP:ATP complex both in the presence and absence of the regulatory protein PaHisZ in order to gain insight into the role of PaHisZ in the allosteric modulation of PaHisG$_S$. We then constructed dynamic cross-correlation matrices (DCCM) for the Cα-atoms during simulations of each system (Supplementary Fig. 10) to analyze the occurrence of correlated motions in the nonactivated and activated dimers for both WT and R56A variants. These plots show a clear shift in the PaHisG$_S$ dimer conformational ensemble upon allosteric activation, with increases in both correlated and anti-correlated motions compared to the nonactivated dimer, and similar effects upon activation of both the wild-type enzyme and the R56A variant. Spearman's rank correlation coefficients between the nonactivated and activated systems are 0.26 for WT-PaHisG$_S$ and 0.37 for R56A-PaHisG$_S$. In contrast, the coefficients between WT-PaHisG$_S$ and R56A-PaHisG$_S$ in each of the nonactivated and activated states are 0.86 and 0.84, respectively. This suggests high similarity between WT-PaHisG$_S$ and R56A-PaHisG$_S$ when the two systems are in the same state (nonactivated *vs.* activated), but low similarity between the nonactivated and activated states of each individual variant. It is expected that external structural perturbations (such as ligand binding or, in this case, the binding of PaHisZ to PaHisG$_S$) would alter such conformational fluctuations, as has been observed in other allosteric systems[36–38], and it can also be seen here that PaHisZ binding increases order in the system.

The conformational behavior of the activated dimer was further explored using the shortest path map (SPM) approach as implemented by Osuna and coworkers[39], which enables the identification of pairs of residues in both the active site and distal positions[40] with the highest contributions to the communication pathways in nonactivated and activated PaHisG$_S$. As described by Osuna[41] and Guo and Zhou[42] in the implementation we have used, the shortest path lengths are computed using the Dijkstra algorithm[43] by going through all nodes of the graph to identify the shortest path from the first to the last protein residue. This allows the identification of which edges of the graph are shorter (i.e. more correlated, see Supplementary Tables 5–7) and more frequently used for going through all residues of the protein. All edges are

then normalized, and only those edges having the largest contributions are represented in the SPM. Comparison of the computed SPM in the nonactivated and activated enzymes (Fig. 5a, b) illustrates that PaHisZ binding increases intermonomer communication pathways across the two subunits of the PaHisG$_S$ dimer, which could be expected to in turn facilitate key interactions between R56 and PRPP across the dimer. This is in overall agreement with the allosteric activation mechanism gleaned from the crystal structures of PaHisG$_S$ and PaATPPRT[29]. Furthermore, when also considering interactions between the dimer and the regulatory protein (Fig. 5c, d), it can be seen that the SPM spans one of the PaHisZ subunits, bridges both monomers of PaHisG$_S$ and communicates with the regulatory protein mainly through helices α7,8 of PaHisZ and β4,5,11 of PaHisG$_S$ (Supplementary Fig. 11). Interestingly, the calculated SPM (Fig. 5d) contains V257, G268 and I269 of the histidine-binding loop of PaHisZ (D256–I269)[19], suggesting a perturbation of the allosteric communication pathway upon histidine binding.

To identify residues that have the largest effect on allosteric signal communication, we have performed node-weakening analysis by removing the nodes corresponding to the residues located at the interface between PaHisG$_S$ and PaHisZ, where the SPM goes through both proteins (5 nodes on PaHisG$_S$ and 7 nodes on PaHisZ), and calculating the change in CPL (characteristic path length) upon removal of each node (Supplementary Table 8)[44–46]. We then selected the three residues displaying the highest impact on the ΔCPL and performed molecular dynamic simulations of four selected *in-silico* mutants of PaATPPRT (Y105A-PaHisG$_S$, Y105F-PaHisG$_S$, N185A-PaHisZ, and K186D-PaHisZ), with the aim of disrupting the communication pathway between PaHisG$_S$ and PaHisZ. When comparing the SPM of the WT-PaATPPRT with those of the various mutants we generated *in-silico*, we see that the pathway is slightly reorganized without displaying critical changes that disrupt the communication signal between the two proteins (Supplementary Fig. 12). These results are in line with a proposal that allosteric activation is due to a global "nesting" effect of PaHisZ over PaHisG$_S$, with a preferred but not unique allosteric activation pathway.

We also analyzed the contribution of each residue to the overall protein dynamics by means of the dynamical flexibility indices (DFI)[47] which measure the dynamic response at each residue in a protein when the system is perturbed (e.g. by a random Brownian kick). This mimics the natural condition in a crowded cellular environment, where the protein is exposed to many different random forces. Thus, the DFI analysis aims to capture the contribution of each position in the protein to the underlying functional dynamics, highlighting the key

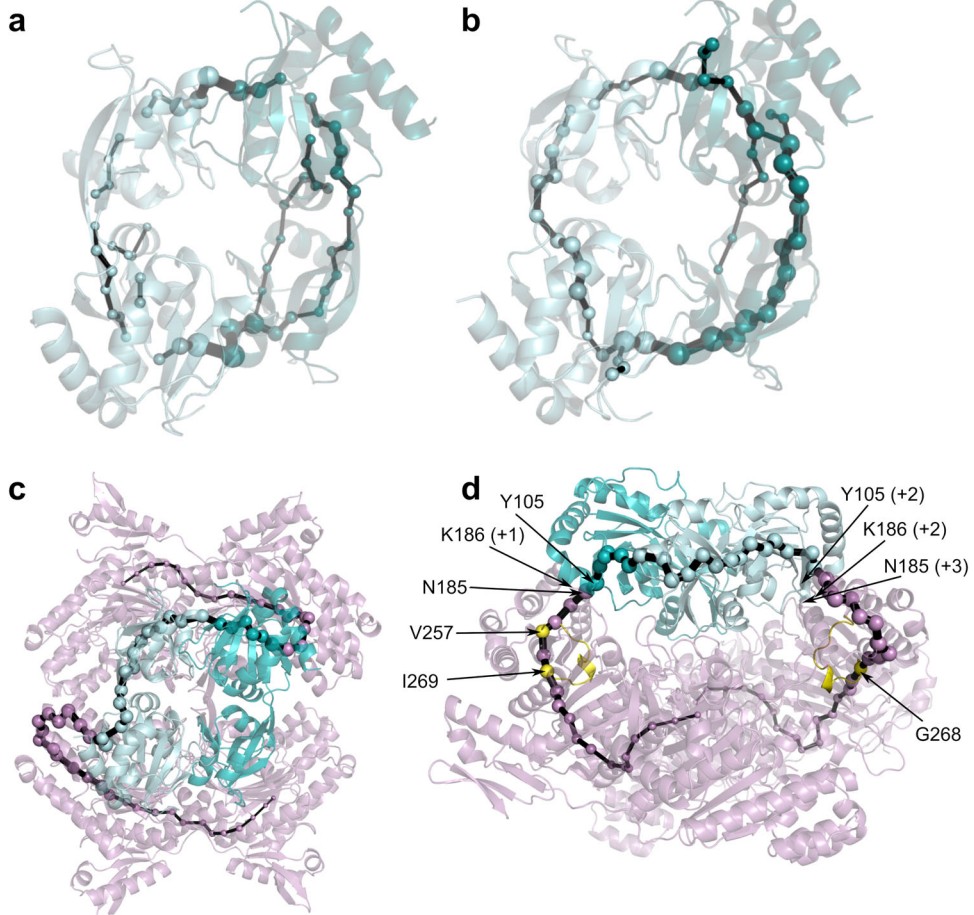

**Fig. 5 | Effect of *Pa*HisZ binding on *Pa*HisG$_S$ dynamic network. a** SPM analysis of nonactivated *Pa*HisG$_S$ dimer. **b** SPM analysis of activated *Pa*HisG$_S$ dimer upon *Pa*HisZ binding (only the catalytic subunits are shown). **c** SPM analysis of *Pa*ATPPRT including *Pa*HisZ residues. **d** Side view (90° rotation) of **c**. *Pa*HisG$_S$ monomers are shown in light and dark teal, while *Pa*HisZ is shown in pink. The histidine-binding loop (D256 – I269) is shown in yellow in **d**. The sizes of the edges (black lines) and vertices (spheres) indicate the strength of the network (the larger size the more pathways available, and thus the higher the importance for allosteric communication). The positions of key residues in the histidine-binding loop and some included in the node-weakening analysis are denoted by arrows; if they are adjacent to the SPM, the number in brackets indicates how many amino-acid residues away they are from the closest residue encompassed by the SPM.

residues and regions that mediate functionally important dynamical information. Our analysis (Supplementary Fig. 13) reveal *Pa*HisZ binding changes the DFI of *Pa*HisG$_S$ compared with nonactivated *Pa*HisG$_S$, in particular at the helices inferred to be important for communication at the *Pa*HisG$_S$-*Pa*HisZ interface from the SPM analysis (Fig. 5), as well as the region containing R32 and R56.

**Insights into the dynamics of allosteric rescue of R56A-*Pa*HisG$_S$**
Additional MD simulations were performed on both nonactivated and activated R56A-*Pa*HisG$_S$:PRPP:ATP complexes to gather knowledge at the atomic level into the allosteric rescue of this variant. Given the catalytic importance of R32 and R56 and their hypothesized involvement in facilitating leaving group departure, we tracked the distance between Cζ of each side chain and Pα of the PP$_i$ moiety of PRPP during our simulations (Fig. 6a). It can be gleaned from the data that the R56 side chain displays a bimodal distribution of distances in WT-*Pa*HisG$_S$ (Fig. 6b). This distribution comprises a peak at ~4.4 Å corresponding to a catalytic conformation in which this side chain forms a salt-bridge with the PRPP PP$_i$ moiety, and another at ~7.8 Å, corresponding to a non-catalytic rotamer of this residue. Binding of *Pa*HisZ shifts the distance distribution towards the catalytically active rotamer (Fig. 6c). This furnishes support for *Pa*HisZ binding constraining the conformational dynamics of *Pa*HisG$_S$, fostering a preorganized active

site in which the R56 guanidinium group is poised to help stabilize the leaving group at the transition state.

Changes in R32 distance distribution upon allosteric activation of WT-*Pa*HisG$_S$ are more subtle, but follow a similar trend as seen for R56, with a peak at ~5.4 Å in WT-*Pa*HisG$_S$ (Fig. 6d) shifting to one at ~4.2 Å in WT-*Pa*ATPPRT (Fig. 6e) to favor interaction with the PP$_i$ moiety. This effect is exacerbated in the absence of R56, where a broad distribution of R32 distances in R56A-*Pa*HisG$_S$ (Fig. 6f) changes to a very narrow peak at ~4.2 Å in R56A-*Pa*ATPPRT (Fig. 6g). These simulations offer a dynamics-based hypothesis for the allosteric rescue of R56A-*Pa*HisG$_S$: in the absence of R56, *Pa*HisZ binding constrains the conformational ensemble of R56A-*Pa*HisG$_S$ mainly to a population where a very narrow distribution of R32 rotamers is sampled that is optimized for a salt-bridge with the PP$_i$ leaving group at the transition state, partially off-setting the loss of such interaction with R56. It is also tempting to speculate that allosteric rescue of R32A-*Pa*HisG$_S$ by *Pa*HisZ might involve in turn a similar constraint of R56 dynamics to facilitate leaving group departure at the transition state and partially compensate for the loss of R32.

**The R32A and R56A substitutions affect the chemical step**
The proposal that allosteric rescue of R56A-*Pa*HisG$_S$, and likely R32A-*Pa*HisG$_S$, by *Pa*HisZ is underpinned by constrained *Pa*HisG$_S$ protein

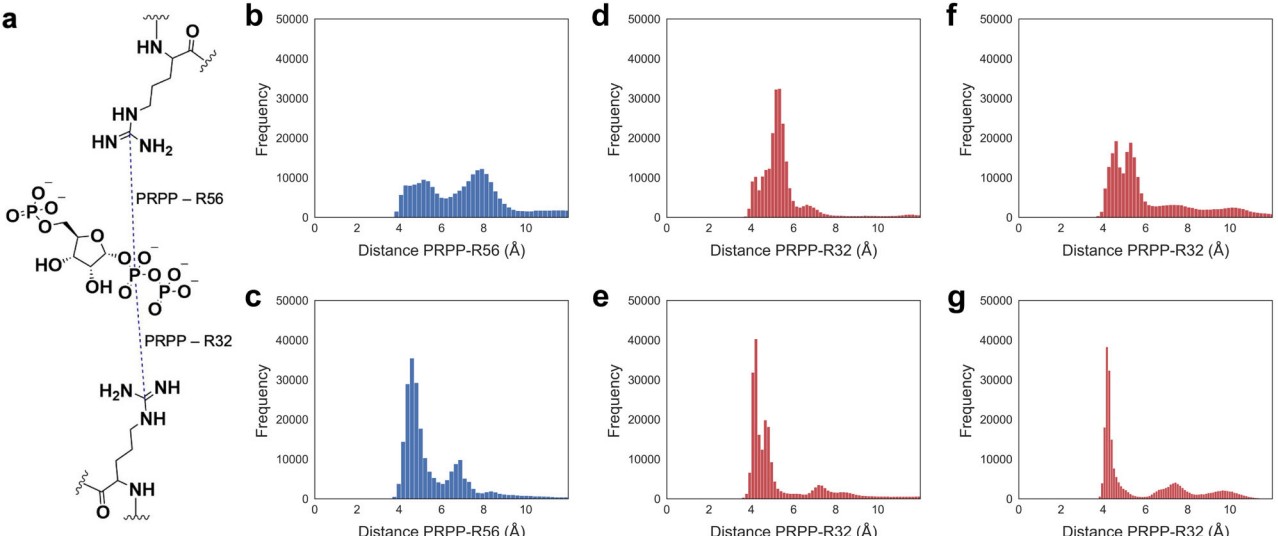

**Fig. 6 | Effect of *Pa*HisZ binding on *Pa*HisG$_S$ R56 and R32 rotamers. a** Schematic rendering of the distances (dashed lines) between Cζ of either R56 or R32 and Pα of the PP$_i$ moiety of PRPP monitored during MD simulations, henceforth referred to as PRPP – R56 distance and PRPP – R32 distance, respectively. **b** Distribution of PRPP – R56 distances in WT-*Pa*HisG$_S$. **c** Distribution of PRPP – R56 distances in WT-*Pa*ATPPRT. **d** Distribution of PRPP – R32 distances in WT-*Pa*HisG$_S$. **e** Distribution of PRPP – R32 distances in WT-*Pa*ATPPRT. **f** Distribution of PRPP – R32 distances in R56A-*Pa*HisG$_S$. **g** Distribution of PRPP – R32 distances in R56A-*Pa*ATPPRT.

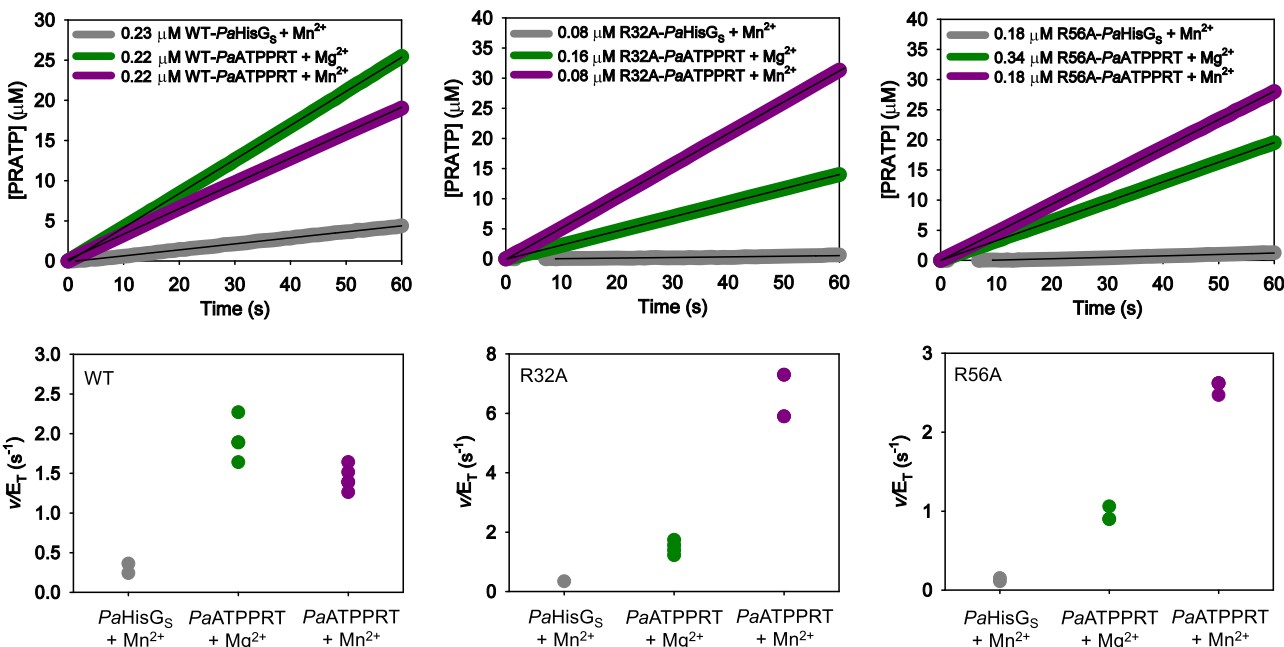

**Fig. 7 | Effect of Mn²⁺ on R32A-*Pa*ATPPRT and R56A-*Pa*ATPPRT reactions. a** PRATP formation time course catalysed by *Pa*HisG$_S$ and *Pa*ATPPRT variants at fixed, saturating concentration of PRPP and ATP. Traces are averages of two to four independent measurements. Black lines are linear regressions of the data. Source data are provided as a Source Data file. **b** Apparent first-order rate constants for *Pa*HisG$_S$ and *Pa*ATPPRT variants. All data points are shown. Two independent measurements were carried out for *Pa*HisG$_S$ variants, and four for *Pa*ATPPRT. For the latter, where less than four data points are apparent, identical rates for two replicates overlap.

dynamics restoring proper active-site electrostatic preorganisation predicts the rescued reaction rates must reflect at least in part the chemical step of the reaction. We have previously demonstrated that replacement of Mg²⁺ by Mn²⁺ increases WT-*Pa*HisG$_S$ $k_{cat}$, and density-functional theory calculations provided a rationale for this effect based on more efficient stabilisation of the negative charges by Mn²⁺ via *d*-orbital bonding to the oxygens of the departing PP$_i$ at the transition state[18]. This was corroborating evidence that chemistry was the rate-limiting step in *Pa*HisG$_S$ catalysis, but not in *Pa*ATPPRT where Mn²⁺ had

no significant impact on $k_{cat}$. This observation is reproduced here. At saturating concentrations of both substrates, Mn²⁺ allows product formation to be detected at a WT-*Pa*HisG$_S$ concentration too low to detect reaction with Mg²⁺, but does not increase the WT-*Pa*ATPPRT reaction rate (Fig. 7). In contrast, when R32A-*Pa*ATPPRT and R56A-*Pa*ATPPRT reactions were carried out with Mn²⁺ instead of Mg²⁺, the rate of product formation increased (Fig. 7a), and the apparent first-order rate constants increased by ~5-fold and ~3-fold, respectively, in comparison with those with Mg²⁺ (Fig. 7b). This indicates the rates of

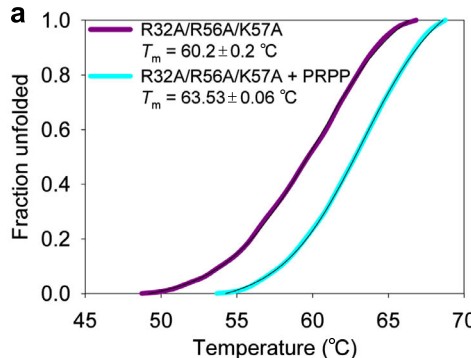
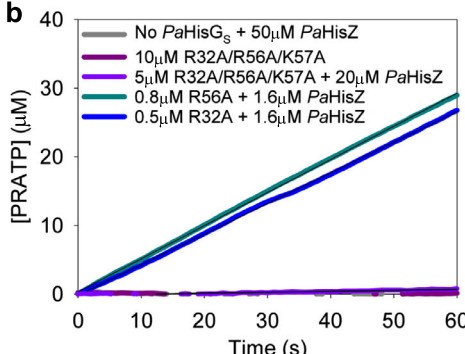

**Fig. 8 | Biochemical characterisation of R32A/R56A/K57A-*Pa*HisG$_S$. a** DSF-based thermal denaturation of R32/R56A/K57A-*Pa*HisG$_S$. Traces are averages of three independent measurements. Lines of best fit to Eq. (2) are in black. Source data are provided as a Source Data file. **b** PRATP formation time course catalysed by *Pa*HisG$_S$ variants. Traces are averages of two independent measurements. Black lines are linear regressions of the data. Source data are provided as a Source Data file.

the rescued mutants reflect the chemical step of the reaction, i.e. unlike WT-*Pa*ATPPRT[18], chemistry is at least partially rate-limiting for R32A-*Pa*ATPPRT and R56A-*Pa*ATPPRT. Interestingly, R32A-*Pa*ATPPRT and R56A-*Pa*ATPPRT apparent first-order rate constants with Mn$^{2+}$ are even higher than the corresponding one for WT-*Pa*ATPPRT. However, these rate constants do not reflect the same reaction step. It is likely that Mn$^{2+}$ enhances the rate of chemistry substantially for WT-*Pa*ATPPRT, but the observed rate constant is dominated by product release[18]. On the other hand, with R32A-*Pa*ATPPRT and R56A-*Pa*ATPPRT, the effect of Mn$^{2+}$ is the enhancement of the rate of chemistry itself. Surprisingly, the apparent first-order rate constants for R32A-*Pa*HisG$_S$ and R56A-*Pa*HisG$_S$ with Mn$^{2+}$ were measurable, $0.12 \pm 0.01$ s$^{-1}$ and $0.13 \pm 0.01$ s$^{-1}$, respectively, only ~2.5-fold lower than that for WT-*Pa*HisG$_S$ with Mn$^{2+}$, although still very low in comparison with those for R32A-*Pa*ATPPRT (~54-fold higher) and R56A-*Pa*ATPPRT (~20-fold higher) (Fig. 7), suggesting the more efficient charge stabilisation ability of Mn$^{2+}$ can partially rescue the activity of these mutants. This observation highlights the importance of electrostatic catalysis in this enzymatic reaction.

Given the proposed role of R56 in stabilising the departure of the negatively charged leaving group, the possibility that a lysine residue could replace R56 with similar catalytic ability was considered. To evaluate this possibility, R56K-*Pa*HisG$_S$ was produced. ESI/TOF-MS analysis resulted in the expected mass (Supplementary Fig. 14a), and DSF showed the mutation does not change the $T_m$ of the protein (Supplementary Fig. 14b). At substrate concentrations saturating for WT-*Pa*HisG$_S$, however, the R56K-*Pa*HisG$_S$ reaction rate is reduced ~24-fold in comparison with the WT-*Pa*HisG$_S$, and although there is an ~11-fold activation in the presence of *Pa*HisZ, the R56K-*Pa*ATPPRT reaction rate is still ~2-fold lower compared with the nonactivated WT variant (Supplementary Fig. 14c). These observations indicate the amino group cannot substitute for the guanidinium group at position 56 of *Pa*HisG$_S$. Furthermore, allosteric activation by *Pa*HisZ cannot rescue catalysis in this case.

### R32A/R56A/K57A-*Pa*HisG$_S$ cannot be rescued by *Pa*HisZ

The hypothesis that R32 and R56 can compensate to a certain extent for the absence of the other in the presence of *Pa*HisZ to restore the electrostatic preorganisation of *Pa*HisG$_S$ active site predicts that removal of both arginine residues would prevent allosteric rescue of catalysis. To test this prediction, the R32A/R56A/K57A-*Pa*HisG$_S$ triple mutant was produced (Supplementary Fig. 1) and ESI/TOF-MS analysis resulted in the expected mass (Supplementary Fig. 2). DSF indicated that the additional mutation does not alter the $T_m$ of the protein (Fig. 8a) as compared with R56A/K57A-*Pa*HisG$_S$ or WT-*Pa*HisG$_S$ $T_m$. Furthermore, PRPP led to an increase in $T_m$, showing the triple mutant

can bind this substrate. As expected, PRATP formation could not be detected with R32A/R56A/K57A-*Pa*HisG$_S$ as catalyst (Fig. 8b). Upon addition of excess *Pa*HisZ, some PRATP formation could be marginally detected above the assay background noise (Fig. 8b), demonstrating R32A/R56A/K57A-*Pa*ATPPRT still retains residual catalytic activity. However, the apparent rate constant is reduced ~777-fold in comparison with that of WT-*Pa*ATPPRT (Supplementary Fig. 6), and ~340-fold and ~222-fold in comparison with those of R32A-*Pa*ATPPRT and R56A-*Pa*ATPPRT, respectively (Supplementary Fig. 15), in accordance with the proposed necessity for at least one of the two arginine residues to aid in leaving group departure for full catalytic power of *Pa*ATPPRT.

## Discussion

Rescue of catalytically impaired enzyme mutants is well established, but not as observed for *Pa*ATPPRT. Chemical rescue by small molecules that mimic missing residue side chains is a useful tool to probe the function of active-site residues in catalysis[48], and a hypernucleophilic cholesterol analogue in which an −OOH group replaces the sterol −OH group could rescue the base-catalysed endoproteolytic activity of a mutant hedgehog protein where the catalytically essential aspartate general base was mutated to an alanine, which rendered the reaction highly impaired with the natural substrate[49]. A catalytically compromised receptor tyrosine kinase carrying mutations in the activation loop tyrosine residues that would otherwise be autophosphorylated could be allosterically rescued by the juxtamembrane segment via autophosphorylation of this segment's Y687, but in this case the effect of the phospho-Y687 is exerted by direct interaction with arginine residues that normally interact with the phosphorylated tyrosine residues in the activation loop[50]. In hetero-tetrameric tryptophan synthase, where catalysis by the β subunit is allosterically activated by the α subunit, the catalytically inactive E109A mutant of the β subunit could not be rescued by the α subunit, but the activity of the E109A-α$_2$β$_2$ complex could be partially restored by CsCl, possibly by modulation of the conformational ensemble of the complex[51]. In human prolyl isomerase CypA, the second-shell S99T mutation, which is highly detrimental to catalysis, can be partially counteracted by additional mutations outside the active site which rescue the dynamics of interconversion between two essential conformations[7].

*Pa*ATPPRT is unique because the rescue of catalytically compromised *Pa*HisG$_S$ mutants by *Pa*HisZ, and even by the orthologous *Ab*HisZ, is truly allosteric since the regulatory subunit binds far from the active site where the mutations exert a detrimental effect on transition state stabilisation. The narrowing of the distribution of states sampled by the *Pa*HisG$_S$ upon *Pa*HisZ binding has a direct impact on the positioning and orientation of R32 and R56, which are better poised to facilitate leaving group departure by electrostatic

stabilisation of the PP$_i$ negative charges. This implies allosteric activation of catalysis in *Pa*ATPPRT involves modulation of the conformational flexibility of the holoenzyme and electrostatic preorganisation of the active site. The catalytic recruitment of R32 and R56 in concert with Mg$^{2+}$ to stabilise PP$_i$ is reminiscent of that of arginine residues in adenylate kinase to promote phosphate transfer from ATP to AMP, where thermally equilibrated protein motions were also proposed to help achieve optimal electrostatic preorganisation[52]. Interestingly, replacement of a key arginine residue for a lysine was detrimental to catalysis in adenylate kinase as well[52].

Another important aspect of the allosterically rescued R32A- and R56A-*Pa*ATPPRT is that $k_{cat}$ is at least partially limited by the chemical step, a drastic change from WT-*Pa*ATPPRT in which $k_{cat}$ is determined by product release[18]. Electrostatic preorganisation exerts its effect on catalysis at the chemical step, i.e. as the reaction progresses from the preorganised Michaelis complex to the transition state. Thus it is paramount that an experimentally measured rate constant purporting to reflect any coupling of protein motions to the preorganisation of the Michaelis complex be limited by the chemical step[9,53]. This is what is observed with the allosteric rescue of the catalytically compromised mutants of *Pa*HisG$_S$, establishing a direct connection between *Pa*HisZ-modulated rotamers of R32 and R56 and electrostatic preorganisation of the active site, which is required for optimal catalysis.

## Methods

### Reagents
All commercially available chemicals were used without further purification. BaseMuncher endonuclease was purchased from AbCam. Ampicillin, dithiothreitol (DTT), isopropyl β-D-1-thiogalactopyranoside (IPTG) and 2-(*N*-morpholino)ethanesulfonic acid-sodium dodecyl sulfate (MES-SDS) were purchased from Formedium. DH5α chemically competent *Escherichia coli*, DpnI were purchased from New England Biolabs (NEB). QIAprep Spin Miniprep, PCR clean-up and Plasmid Midi kits were from Qiagen. Ethylenediaminetetraacetic acid (EDTA)-free Complete protease inhibitor cocktail was from Roche. ATP, C43(DE3) and BL21(DE3) chemically competent *E. coli*, D$_2$O, glycerol, histidine, imidazole, lysozyme, PRPP, potassium chloride, and tricine were purchased from Sigma-Aldrich. Agarose, dNTPSs, kanamycin, 4-(2-hydroxyethyl)piperazine-1-ethanesulfonic acid (HEPES), MgCl$_2$, NaCl, PageRuler Plus Prestained protein ladder, PageRuler™ Plus Prestained protein ladder, and SYPRO orange protein gel stain were from ThermoFisher Scientific. DNA oligonucleotide primers were synthesised by Integrated DNA technologies (IDT).

### Site-directed mutagenesis of *Pa*HisG$_S$
Site-directed mutagenesis was carried out with overlapping primers according to the method of Liu and Naismith[54]. Primer sequences are listed in Supplementary Table 9. For the triple mutant R32A/R56A/K57A-*Pa*HisG$_S$, the R56A/K57A-*Pa*HisG$_S$ expression vector was used as DNA template. For all other mutants, WT-*Pa*HisG$_S$ expression vector was used. Correct insertion of each mutation was confirmed by DNA sequencing performed by either Eurofins Genomics or DNA Sequencing & Services at University of Dundee.

### Protein expression and purification
*Pa*HisG$_S$, *Pa*HisZ, *Mycobacterium tuberculosis* pyrophosphatase (*Mt*PPase), and tobacco etch virus protease (TEVP) were produced as previously described[28]. *Ab*HisZ was produced as previously reported[26]. All *Pa*HisG$_S$ mutants were expressed and purified by the same protocol as *Pa*HisG$_S$[28]. His-tagged *Pa*HisZ was purified by the same protocol as *Pa*HisZ up to and including the first chromatography[28], after which fractions producing a single band at the expected MW in SDS-PAGE were pooled and dialysed against 2 × 2L of 20 mM HEPES pH 8.0, concentrated using 10,000 molecular weight cutoff (MWCO)

ultrafiltration membranes (Millipore), aliquoted and stored at −80 °C. His-tagged *Pa*HisZ, C115-, C115S-, D179A-, D179-, R32A-, R56A-, R56A/K57A-, R56K, and R32A/R56A/K57A-*Pa*HisG had their intact mass determined by ESI/TOF-MS, and D179N-*Pa*HisG$_S$ tryptic peptides underwent MS/MS analysis to confirm the mutation, all performed by the BSRC Mass-Spectrometry and Proteomics Facility at the University of St Andrews. The concentration of WT-*Pa*HisG$_S$, WT-*Pa*HisZ, *Mt*PPase and TEVP was determined as published[28]. The concentration of His-tagged *Pa*HisZ and *Pa*HisG$_S$ mutants was determined spectrophotometrically (NanoDrop) at 280 nm based on the theoretical extinction coefficients (ε$_{280}$) calculated in the ProtParam tool of Expasy: ε$_{280}$ of 27,930 M$^{-1}$ cm$^{-1}$ for His-tagged *Pa*HisZ; 8940 M$^{-1}$ cm$^{-1}$ for all *Pa*HisG$_S$ mutants.

### DSF
DSF measurements (λ$_{ex}$ = 490 nm, λ$_{em}$ = 610 nm) were performed in 96-well plates on a Stratagene Mx3005p instrument. Reactions (50 μL) contained 100 mM tricine, 100 mM KCl, 15 mM MgCl$_2$, 4 mM DTT pH 8.5, 7 μM enzyme, and either 0 mM or 2 mM PRPP, with 5X Sypro Orange (Invitrogen) added to each well. Thermal denaturation curves were recorded over a temperature range of 25–93 °C with increments of 1 °C min$^{-1}$. Control curves lacked protein and were subtracted from curves containing protein. All measurements were carried out in triplicate.

### Analytical size-exclusion chromatography
Analytical size-exclusion chromatography was performed on a Superdex 200 10/300 GL column (GE Healthcare) attached to a Bio-Rad NGC FPLC at 4 °C. WT-, C115S-, R32A-, R56A- and R56/K57A-*Pa*HisG$_S$ (1 mg mL$^{-1}$, pre-incubated with DTT [2 mM]) were loaded onto the column (equilibrated with 20 mM HEPES pH 8.0) and eluted with 1 column volume of 20 mM HEPES pH 8.0 at 0.225 mL min$^{-1}$.

### Enzyme activity assay
Unless stated otherwise, all reactions (500 μL) were carried out in 1 cm path-length quartz cuvettes under initial-rate conditions at 20 °C, and the increase in absorbance at 290 nm due to the formation of PRATP (ε$_{290}$ = 3600 M$^{-1}$ cm$^{-1}$)[35] was monitored for 60 s in a Shimadzu UV-2600 spectrophotometer in 100 mM tricine, 100 mM KCl, 15 mM MgCl$_2$, 4 mM DTT, pH 8.5, 10 μM *Mt*PPase, and various concentrations of PRPP and ATP. Cuvettes were incubated in the spectrophotometer at 20 °C for 3 min before reaction was initiated by the addition of PRPP. Control reactions lacked enzyme.

### Activity of *Pa*HisG$_S$ mutants in the absence of *Pa*HisZ
Activity of C115A-, C115S-, D179A-, D179N-, R32A-, R56A-, and R56A/K57A-*Pa*HisG (5 μM) was assayed for 60 s in the presence of 5.6 mM ATP and 2 mM PRPP. WT-*Pa*HisG$_S$ (5 μM) was included in as a positive control. Additionally, the activity of C115S-, R32A-, R56A- and R56A/K57A-*Pa*HisG (10 μM) was assayed for 870 s in the presence of 5.6 mM ATP and 2 mM PRPP. Exceptionally, activity of R56K-*Pa*HisG (6 μM) was assayed both in the absence and presence of *Pa*HisZ (20 μM).

### Determination of apparent $K_D$ for *Pa*HisZ, His-tagged *Pa*HisZ and *Ab*HisZ
Initial velocities were measured in 5.6 mM ATP and 2 mM PRPP. For *Pa*HisZ $K_D$, 0.42 μM WT-, 0.59 μM C115A-, 1.1 μM R32A-, 1.1 μM R56A- and 1.7 μM R56A/K57A-*Pa*HisG$_S$ were assayed in the presence of varying concentrations of *Pa*HisZ (0–5.1 μM for WT-*Pa*HisG$_S$; 0–15 μM for C115S-*Pa*HisG$_S$; 0–8 μM for R32A-*Pa*HisG$_S$; 0–15 μM for R56A-*Pa*HisG$_S$; and 0–11 μM for R56A/K57A-*Pa*HisG$_S$). For His-tagged *Pa*HisZ $K_D$, 0.23 μM WT-, 0.49 μM R32A-, and 0.79 μM R56A-*Pa*HisG$_S$ were assayed in the presence of varying concentrations of His-tagged-*Pa*HisZ (0–1.6 μM). For *Ab*HisZ $K_D$, 0.19 μM WT- and 2.5 μM R56A-*Pa*HisG$_S$ were assayed in the presence of varying concentration of *Ab*HisZ (0–19.7 μM

for WT-*Pa*HisG$_S$; 0–26.9 μM for R56A-*Pa*HisG$_S$). Alternatively, 1 μM C115S-, 1 μM R32A-, 1 μM R56A-, and 1.6 μM R56A/K57A-*Pa*HisG$_S$ were assayed in the presence of 20 μM BSA.

### WT-, C115A-, D179A- and D179N-*Pa*HisG$_S$ saturation kinetics

Initial rates for 3.4 μM WT-, 10.1 μM C115A-, 9.2 μM D179A- and 10.0 μM D179N-*Pa*HisG$_S$ were measured at saturating concentrations of one substrate (either 2 mM PRPP or 5.6 mM ATP) and varying concentrations of the other, either ATP (0–5.6 mM) or PRPP (0–2 mM).

### C115S-, R32A-, R56A- and R56A/K57A-*Pa*ATPPRT saturation kinetics

Initial rates for 0.4 μM WT-, 0.5 μM C115S-, 0.9 μM R32A-, 0.9 μM R56A-, and 1.2 μM R56A/K57A-*Pa*ATPPRT were measured at saturating concentrations of one substrate (either 2 mM PRPP or 5.6 mM ATP) and varying concentrations of the other, either ATP (0–5.6 mM) or PRPP (0–2 mM). Alternatively, initial rates for 0.22 μM WT-*Pa*ATPPRT with His-tagged *Pa*HisZ replacing *Pa*HisZ were measured at saturating concentration of one substrate (wither 2 mM PRPP or 2.8 mM ATP) and varying concentrations of the other, either ATP (0–2.8 mM) or PRPP (0–2 mM).

### WT- and R56A-*Pa*HisG$_S$/*Ab*HisZ saturation kinetics

Initial rates for 0.26 μM WT-, and 2.5 μM R56A-*Pa*HisG$_S$/*Ab*HisZ (the latter was assumed from the concentrations of 2.5 μM R56A-*Pa*HisG$_S$ and 19.7 μM *Ab*HisZ) were measured at saturating concentrations of one substrate (either 2 mM PRPP or 2.8 mM ATP for WT- and 5.6 mM for R56A-*Pa*HisG$_S$/*Ab*HisZ) and varying concentrations of the other, either ATP (0–2.8 mM for WT-; 0–5.6 mM for R56A-*Pa*HisG$_S$/*Ab*HisZ) or PRPP (0–2 mM). For WT-*Pa*HisG$_S$/*Ab*HisZ, background rates of control reactions lacking *Ab*HisZ were subtracted.

### Inhibition by histidine

Initial rates for 0.43 μM C115S-, 0.49 μM R32A-, 0.79 μM R56A-, and 2.6 μM R56A/K57A-*Pa*HisG$_S$ with either 3.2 μM His-tagged *Pa*HisZ (for C115S-, R32A-, and R56A-*Pa*HisG$_S$) or 6.4 μM His-tagged *Pa*HisZ (for R56A/K57A-*Pa*HisG$_S$) were measured at saturating concentrations of both substrates (2 mM PRPP and 5.6 mM ATP) in the presence and absence of 1 mM histidine. Initial rates for 0.26 μM WT-, and 2.5 μM R56A-*Pa*HisG$_S$/*Ab*HisZ (the latter was assumed from the concentrations of 2.5 μM R56A-*Pa*HisG$_S$ and 19.7 μM *Ab*HisZ) were measured at saturating concentrations of both substrates (2 mM PRPP and either 2.8 mM ATP for WT- or 5.6 mM for R56A-*Pa*HisG$_S$/*Ab*HisZ) in the presence and absence of 1 mM histidine.

### $^{31}$P-NMR spectra of the *Pa*HisG$_S$ and *Pa*ATPPRT reactions

In 500 μL reactions, either 10 μM R56A-, 10 μM C115S-, 20 μM R32A- or 20 μM R56A/K57A-*Pa*HisG$_S$ was incubated in the presence or absence of 30 μM *Pa*HisZ in 100 mM tricine, 100 mM KCl, 15 mM MgCl$_2$, 4 mM DTT, pH 8.5, 20 μM *Mt*PPase, 2 mM PRPP and 5.6 mM ATP for 1 h at 20 °C. Proteins were removed by passage through 10,000 MWCO Vivaspin centrifugal concentrators, after which 100 μL of D$_2$O was added to each sample. $^{31}$P-NMR spectra were recorded on either a Bruker AV 400 or Bruker AVII 400 spectrometer, and a total of 128 scans were collected for each sample.

### Activity of R32A/R56A/K57A-*Pa*HisG$_S$

R32A/R56A/K57A-*Pa*HisG$_S$ (10 μM in the absence of His-tagged *Pa*HisZ and 5 μM in the presence of 20 μM His-tagged *Pa*HisZ) was assayed for catalytic activity under initial-rate conditions in the presence of 5.6 mM ATP and 2 mM PRPP. R32A- and R56A-*Pa*HisG$_S$ (0.5 μM and 0.8 μM, respectively) in the presence of 1.6 μM His-tagged *Pa*HisZ were assayed as positive controls for allosteric rescue. Negative controls contained 50 μM His-tagged *Pa*HisZ but lacked *Pa*HisG$_S$.

### WT-, R32A- and R56A-*Pa*ATPPRT activities with Mn$^{2+}$

Initial rates were determined for 0.23 μM WT-*Pa*HisG$_S$, 0.22 μM WT-*Pa*ATPPRT, 0.08 μM R32A-*Pa*HisG$_S$, 0.08 μM R32A-*Pa*ATPPRT, 0.18 μM R56A-*Pa*HisG$_S$, and 0.18 μM R56A-*Pa*ATPPRT in 100 mM tricine, 100 mM KCl, 15 mM MnCl$_2$, 4 mM DTT, pH 8.5 and 10 μM *Mt*PPase at saturating concentrations of ATP (1.4 mM for WT enzymes; 5.6 mM for mutant enzymes) and PRPP (1 mM for WT enzymes; 2 mM for mutant enzymes). Initial rates were also determined for 0.22 μM WT-*Pa*ATPPRT, 0.16 μM R32A-*Pa*ATPPRT, and 0.34 μM R56A-*Pa*ATPPRT in 100 mM tricine, 100 mM KCl, 15 mM MgCl$_2$, 4 mM DTT, pH 8.5 and 10 μM *Mt*PPase at saturating concentrations of ATP (5.6 mM) and PRPP (2 mM).

### Crystallisation, X-ray data collection and data processing

Crystals of R56A-*Pa*HisG$_S$ were grown, soaked in PRPP and ATP and stored as described for WT-*Pa*HisG$_S$[29], whereas crystals of R56A-*Pa*ATPPRT were grown as described for WT-*Pa*ATPPRT[28] and soaked in PRPP and ATP and stored as described for WT-*Pa*ATPPRT[29]. X-ray diffraction data for R56A-*Pa*HisG$_S$ were collected in house as previously reported[28] and processed with iMosflm[55], while data for R56A-*Pa*ATPPRT were collected at Diamond Light Source (UK) and processed at the automated processing pipeline at Diamond with Xia2[56] integrated with DIALS[57]. R56A-*Pa*HisG$_S$ and R56A-*Pa*ATPPRT structures were solved by molecular replacement in MOLREP using WT-*Pa*HisG$_S$:PRPP:ATP (PDB ID: 6FCT) and WT-*Pa*ATPPRT:PRPP:ATP (PDB ID: 6FU2)[29] structures, respectively, as search models. Structures were refined using cycles of model building with COOT[58] and refinement with Refmac[59]. ATP was modelled at either 70% or 80% occupancy in R56A-*Pa*ATPPRT.

### MD simulations

Molecular dynamics simulations were performed on WT-*Pa*HisG$_S$:PRPP:ATP (PDB ID: 6FCT)[29], WT-*Pa*ATPPRT:PRPP:ATP (PDB ID: 6FU2)[29], R56A-*Pa*HisG$_S$:PRPP:ATP (PDB ID: 7Z8U), and R56A-*Pa*ATPPRT:PRPP:ATP (PDB ID: 7Z6R). Any missing regions in the structures were reconstructed using Modeller v. 9.23[60]. In the case of *Pa*HisZ, these were reconstructed using the lowest energy conformation prediction combined with visual inspection; in other systems, *Pa*HisG$_S$ was used as a template (as well as for modeling the position of the binding site magnesium ions). A distal extra Mg$^{2+}$ was deleted from R56A-*Pa*HisG$_S$:PRPP:ATP structure. Finally, the adenine moiety of ATP was flipped into a catalytically productive conformation in all the starting structures. Protonation states of all titratable residues were determined based on a combination of empirical screening using PROPKA v3.1[61], and visual inspection of the local environment. The E122, E163 and H103 side chains were predicted to be found in their ionized states. Previously prepared WT-*Pa*ATPPRT:PRPP:ATP was used as starting point to introduce, in silico, the point mutations from the node-weakening analysis: Y105A-*Pa*HisG$_S$, Y105F-*Pa*HisG$_S$, N185A-*Pa*HisZ, and K186D-*Pa*HisZ. Corresponding rotamers were selected using the Dunbrack 2010 Rotamer Library[62].

Partial charges for the ligand PRPP were calculated *in vacuo* at the HF/6-31G* level of theory using Gaussian 16 Rev. A.03[63], and fitted using the standard restrained electrostatic potential (RESP) protocol as implemented in Antechamber[64] (Supplementary Table 10). All other force field terms for PRPP were then described using the Amber force field ff14SB[65] together with revised parameters to describe bioorganic phosphates[66]. The parameters for ATP were taken from the literature[67]. We used an octahedral cationic dummy model to describe Mg$^{2+}$, following from previous successful results using this model[68,69].

All MD simulations were performed using the GPU-accelerated version of Amber16[70], with the protein and water molecules described

using the amber force field ff14SB[65] and the TIP3P[71] water model, respectively. All systems were solvated in an octahedral box of water molecules, extended 8 Å from the closest solute molecule in every direction. Each system was neutralized by adding $Na^+$ or $Cl^-$ counterions to ensure overall charge neutrality. Counterions were placed using the "addions" approach as implemented in Amber16[70]. The dimeric and hetero-octameric forms of the enzyme were simulated for $10 \times 500$ ns and $5 \times 500$ ns, respectively, in the NPT ensemble. The solvated systems were first minimized using 5000 steps of steepest descent minimization with 500 kcal mol$^{-1}$ Å$^{-2}$ positional restraints placed on all solute atoms to minimize all hydrogen atoms and solvent molecules, followed by 5000 steps of conjugate gradient minimization, with the restraint dropped to 5 kcal mol$^{-1}$ Å$^{-2}$. The minimized system was then heated from 0 to 300 K in an NVT ensemble over 250ps of simulation time using the Berendsen thermostat[72] with a time constant of 1ps for the coupling while maintaining the 5 kcal mol$^{-1}$ Å$^{-2}$ restraint. The restraint was then limited to heavy atoms of the substrates for a further 200ps of NPT equilibration, followed by 200ps of unrestrained equilibration. After minimization, five distance restraints were applied during the simulations to PRPP and ATP (four to PRPP and one to ATP) to prevent dissociation of the substrates from the active site throughout the MD simulations (see Supplementary Table 11 for a full list of the restraints applied). Note that no restraints were applied between the substrate and any of the regions of interest to ensure full conformational freedom of such regions.

All production-quality simulations were performed using a 2 fs time step, with the SHAKE algorithm[73] used to constrain all bonds containing hydrogen atoms. Temperature and pressure were controlled by the Langevin thermostat with a collision frequency of 1ps$^{-1}$[174], and the Berendsen barostat with a 1 ps coupling constant[72]. A cutoff of 8 Å was applied to all non-bonded interactions, with the long range electrostatics being evaluated using the particle mesh Ewald (PME) approach[75]. The root mean square deviations (RMSD) of all backbone atoms for each system during the production runs is shown in Supplementary Fig. 16. Unless stated otherwise, all analysis was performed using CPPTRJ[76]. Principal Component Analysis was performed in Cartesian coordinate space on the Cα atoms of the shared dimeric region of all the studied systems, by first root-mean-square fitting all the trajectories to the WT-$Pa$HisG$_S$ crystal structure. DCCMs were generated with Bio3D[77]. Two-sided Spearman's rank order correlation tests were used to measure the monotonicity of the relationship between distinct correlation matrices. The upper triangle of each $416 \times 416$ matrix was used, representing a sample size of $n = 86,528$. The standard assumptions (data measured on an interval scale, and the two variables are monotonically related) were used.

### Kinetics and thermal denaturation data analysis

Kinetic and thermal denaturation data were analysed by the nonlinear regression function of SigmaPlot 14.0 (SPSS Inc.). Data points with error bars represent mean±SEM of two to four independent measurements, and kinetic and equilibrium constants are given as mean±fitting error. Alternatively, all data points were plotted. Substrate saturation curves at a fixed concentration of the co-substrate were fitted to Eq. (1). Thermal denaturation data were fitted to Eq. (2). Initial rate data at varying concentrations of HisZ were fitted to either Eqs. (3) or (5). The concentration of ATPPRT at any concentration of $Pa$HisG$_S$ and either $Pa$HisZ or $Ab$HisZ was calculated according to Eq. (4). In Eqs. $1 - 5$, $k_{cat}$ is the steady-state turnover number, $v$ is the initial rate, $E_T$ is total enzyme concentration, $K_M$ is the apparent Michaelis constant, $S$ is the concentration of the varying substrate, $V_{max}$ is the maximal velocity, $F_U$ is fraction unfolded, $T$ is the temperature in °C, $T_m$ is the melting temperature, $c$ is the slope of the transition region,

and $LL$ and $UL$ are folded and unfolded baselines, respectively, $h$ is the Hill coefficient, $K_{0.5}$ is the concentration of $Ab$HisZ at the inflection point, $G$ is the concentration of $Pa$HisG$_S$, $Z$ is the concentration of either $Pa$HisZ or $Ab$HisZ, $K_D^{app}$ is the apparent equilibrium dissociation constant, and $ATPPRT$ is the concentration of either $Pa$ATPPRT of $Pa$HisG$_S$/$Ab$HisZ complex.

$$\frac{v}{E_T} = \frac{k_{cat}S}{K_M + S} \quad (1)$$

$$F_U = LL + \frac{UL - LL}{1 + e^{(T_m - T)/c}} \quad (2)$$

$$v = V_{max} \frac{G + Z + K_D^{app} - \sqrt{\left(G + Z + K_D^{app}\right)^2 - 4GZ}}{2G} \quad (3)$$

$$ATPPRT = \frac{G + Z + K_D^{app} - \sqrt{\left(G + Z + K_D^{app}\right)^2 - 4GZ}}{2} \quad (4)$$

$$v = \frac{V_{max}Z^h}{K_{0.5} + Z^h} \quad (5)$$

### Reporting summary

Further information on research design is available in the Nature Portfolio Reporting Summary linked to this article.

### Data availability

Structure factor amplitudes and coordinates for the crystal structures of R56A-$Pa$HisG$_S$:PRPP:ATP and R56A-$Pa$ATPPRT:PRPP:ATP were deposited to the Protein Data Bank under accession numbers 7Z8U and 7Z6R, respectively. MD simulations were based on coordinates for WT-$Pa$HisG$_S$:PRPP:ATP and WT-$Pa$ATPPRT:PRPP:ATP downloaded from the Protein Data Bank under accession numbers 6FCT and 6FU2, respectively. All protein mass spectrometry data were deposited to FigShare under DOIs https://doi.org/10.6084/m9.figshare.19658367 (D179N-$Pa$HisG$_S$ tryptic digestion analysis) and https://doi.org/10.6084/m9.figshare.19658229 (intact mass analysis for remaining $Pa$HisG$_S$ variants). Parameters used to describe the ligands and the magnesium dummy model, input files, starting structures, topologies and snapshots from our molecular dynamics simulations are available for download from Zenodo under https://doi.org/10.5281/zenodo.7077771 [https://zenodo.org/record/7077771#.Y2pZReTP1D8]. Source data are provided with this paper.

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

## Acknowledgements

This work was supported by the Biotechnology and Biological Sciences Research Council (BBSRC) [Grant BB/M010996/1] via EASTBIO Doctoral Training Partnership studentships to B. J. R. and G. F., by Stiftelsen Olle Engkvist Byggmästare [Grant 190-0335] and the Knut and Alice Wallenberg Foundation [Grants 2018.0140 and 2019.0431] to S.C.L.K., and by the European Union's Horizon 2020 Research and Innovation Programme *via* a Marie Sklodowska-Curie fellowship [Grant 890562] to M.C. The simulations were enabled by resources provided by the Swedish National Infrastructure for Supercomputing (SNIC, UPPMAX), partially funded by the Swedish Research Council [Grant 2016-07213]. X-ray diffraction data from R56A-*Pa*ATPPRT crystals were collected at Diamond Light Source in the UK. The authors thank Dr Huanting Liu of the University of St Andrews for advice on site-directed mutagenesis.

## Author contributions

G.F. carried out all experimental work except where noted, and wrote the manuscript. B.J.R. expressed and purified *Ab*HisZ and carried out the sequence comparison with *Pa*HisZ. J.N. expressed and purified R32/R56A/K57A-*Pa*HisG$_S$. M.S.A. supervised the protein crystallography work. M.C. carried out the computational chemistry work and wrote the manuscript. S.C.L.K. supervised the computational chemistry work and wrote the manuscript. R.G.d.S. conceived and supervised the research and wrote the manuscript. All authors analysed data.

## Competing interests

The authors declare no competing interests.
