## [Peer Review File · Nature Communications]

Allosteric rescue of catalytically impaired ATP phosphoribosyltransferase variants links protein dynamics to active-site electrostatic preorganisationReviewers' Comments:

Reviewer #1:

Remarks to the Author:

ATP phosphoribosyltransferase (ATPPRT) catalyzes the first step in histidine biosynthesis and is of great interest for histidine production as well as antimicrobial drug discovery. Of particular interest in the current study, it is a multimeric enzyme composed catalytic (HisGS) and regulatory (HisZ) subunits. Although HisGS alone is functional, it is 29-fold less efficient than the holoenzyme. Here, Fisher et al. have shown that without the regulatory subunit, certain active-site mutations, especially of Arg32 and Arg56, are detrimental to activity; however, addition of HisZ from either of two orthologues resulted in recovery of most of the lost activity, demonstrating the critical role of the regulatory subunit in establishing a productive conformation in its catalytic neighbour. While crystal structures of the Arg56Ala mutants of HisGS and the holoenzyme soaked with ATP and PRPP provided no structural features that could explain the functional differences between these two mutant complexes, MD simulations revealed significant changes. Of great importance, simulations of the corresponding wild-type enzymes highlighted alterations in the overall dynamics and in the motion of the two Arg residues that contact the pyrophosphate (PPi) leaving group of PRPP under the influence of the regulatory subunit. Specifically, the holoenzyme had fewer correlated and anti-correlated motions and exhibited much narrower distributions of distances between PPi and the two flanking Arg residues. The combination of steady-state kinetics and MD simulation presented in this study provides an excellent account for how the altered dynamics of the holoenzyme translates to much greater catalytic efficiencies in all variants tested.

Although allostery in multimeric proteins has been studied for many decades, the mechanistic details are often greatly lacking. The current investigation has removed much of this mystery, at least for ATPPRT, and it will make a great example for future comparisons with other multimeric allosteric enzymes. I support publication after the authors address the following items.

1. The authors note that while the WT holoenzyme shows little dependence on Mg vs Mn, the two Arg mutants show enhanced catalysis with Mn (Fig. 7). Their interpretation is that this shows that chemistry is rate contributing in the mutants but not WT. What they failed to identify, however, is the surprising outcome that the two mutants are substantially better catalysts than WT in the presence of Mn. In fact, R32A is 4–5-fold more active. The authors should make an attempt to explain the observation.

2. On p. 20, it is stated, "these simulations reinforce the proposed role of R32 in stabilization of the leaving group at the transition state." If I've understood the simulations properly, they were performed on the Michaelis complexes and therefore any stabilization identified in these experiments is relevant in the ground state. Perhaps these interactions become even more important during the chemical step, but such information does not come from these experiments. I think some careful rewording should clarify the authors' intent.

The remaining comments are primarily typographical errors.

3. In the abstract, delete the hyphen between 20 and Å. Also, an en dash should be used instead of a colon for "protein–protein interactions".

4. On p. 4, delete the hyphen between ribosyl and 1. Like other sugar phosphates, there are two separate words.

5. On p. 4, delete the comma between holoenzyme and allosterically. Commas should generally not separate the subject and predicate.

6. On p. 6, "structure" should be added after "PaATPPRT" at the end of the first sentence.

7. On p. 6, it states, "the multiple-turnover pre-steady-state rate-constant was in agreement with k_{cat} , suggesting chemistry is rate-limiting in the nonactivated enzyme reaction." Add a comma between "multiple-turnover" and "pre-steady-state" (they are coordinate adjectives); remove the

hyphen in "rate-constant"; and remove the hyphen in "rate-limiting" (a hyphen is only used when these two words function as a compound modifier immediately preceding a noun/noun phrase, as in "rate-limiting step"). Rate-limiting also appears at the end of the subsequent sentence in this paragraph.

8. On p. 7, "molecular mass" should be "molecular masses", as there are multiple proteins, each with a unique mass. Correspondingly, at the end of the same sentence, change "predicted value" to "predicted values".

9. On p. 7, the last sentence of the first paragraph is complex (it contains three main clauses). I recommend splitting into two, beginning the second with "PRATP formation was readily..."

10. On p. 7, second paragraph, it is stated, "The kcat for the mutants were reduced..." kcat is singular, so in order to agree with "were", a plural term needs to be added, such as "kcat values".

11. In Fig. 2c, the vertical scale is in units of $\mu\text{M s}^{-1}$, which is a rate. Therefore, it should be v, not [PRATP]. The same is true in Fig. 3a.

12. In the legend to Fig. 2d, change "Close-up" to "Close-ups".

13. On p. 13, last sentence, "thus" is an adverb, not a conjunction, so it requires a semicolon before it (not a comma), and a comma after it.

14. In Fig. 3c legend, provide the PDB code for the crystal structure, even though it may appear elsewhere in the document.

15. In several locations (listed below), the definite article is used when it ordinarily would not and should be deleted.

a. p. 15, penultimate line, "absent in the R56A-PaHisGS (Fig. 4)."

b. p. 17, last line, "the region containing the R32 and R56."

c. Fig. 6 legend, "between C ζ of either R56 or R32 and the Pa of the PPI moiety." Here, "the" could be used but then it should appear before C ζ for consistency. Personally, I would not use "the" because Pa is unambiguous. Whatever option is chosen should be made consistent with the text that appears near the end of the first sentence on p. 19.

16. In Fig. 4 legend, it should read, "The A56 side chain is contributed by one of the adjacent PaHisGS subunits. In the next line, Mg²⁺ is depicted as "a sphere" (add the indefinite article).

17. On p. 17, one of the sentences begins, "Interestingly, the calculated SPM (Fig. 6)", but Fig. 6 does not contain any SPMs. Perhaps Fig. 5d was intended. In the next sentence, DFI is defined as "dynamical flexibility indexes" but in the supplementary information, "indices" is used instead. Choose one and use it consistently.

18. On p. 19, 3rd and 4th sentences read, "It can be gleaned from the data that the R56 side chain displays a bimodal distribution of distances in WT-PaHisGS (Fig. 6b). These comprise..." The plural pronoun "these" has not been properly defined. The authors are referring to what appears to be the two distances of maximal frequency in the bimodal distribution, but these positions have not yet been identified in the text. Otherwise, the only plural term in the preceding sentence is "distances", but in that context, it is the entire domain, not just the two points, that is being described.

19. In Fig. 6a, the two arginines are rendered as free amino acids, but they should be residues within a peptide chain. Also, the Å symbol has no meaning here (the adjacent figures already provide the unit).

20. On p. 21, the apparent first-order rate constants with Mn²⁺ are compared with those for R32A and R56A holoenzymes, which are both larger. However, in parentheses, the values are described as being higher in one case and lower in the other.

21. On p. 25, in the first appearance of an organism's name, the genus should be provided in full. Thus, it should be *Escherichia coli*.

22. The protease inhibitor cocktail has been written with a Scandinavian letter ø, but this is not correct. Either write it as Complete or cComplete, which is how Roche stylizes it.

23. On p. 26, first line, tricine is the last item in the list and therefore requires "and" before it and no comma after it.

24. On p. 30, it should be "after which 100 μL of D₂O was added" (singular verb form). In the next sentence, change "spectrophotometer" to "spectrometer".

25. On p. 32, first line, delete the comma after PRPP.
26. On p. 34, in eqs 3 and 4, the first set of brackets should be removed, as only addition is being performed. When comparing the radicands, the two sets of brackets are different sizes; perhaps typed parentheses were used in eq 3, while brackets were used in eq 4.
27. On p. 42, "Conceived" should be lowercase.

Reviewer #2:

Remarks to the Author:

This is a detailed and well written manuscript describing allostery of an important biosynthetic enzyme. ATPPRT catalyses the first step of His biosynthesis and the authors here explore the role of specific residues in catalysis by mutation - all of which are detrimental, but the catalytic impairment is overcome, to some extent by the inclusion of the allosteric domain non-covalent partner, HisZ. I have really enjoyed this manuscript, but I vacillate a bit between characterising this allosteric rescue as expected or exciting.

These authors previous works have proposed a mechanism for the activation of the catalytic domains by HisZ, organisation of charged functionality to facilitate loss of diphosphate leaving group - this is established. Is it therefore that surprising that significantly impaired mutants in the catalytic site, also are activated by HisZ? It feels a little as if the activations maybe somewhat overstated because of the very low rates of catalysis without HisZ present. Inclusion of the HisZ does improve the rates, but there is still impairment, relative to the wild-type enzyme. As is to be expected.

Also while the allosteric rescue of the R32 and R56 mutants makes sense in terms of the proposed mechanism for the HisZ activation of the enzyme, the activation fold is most significant for the C115S mutant. This results does not seem to be commented on at length.

The data is extensive and this helps this manuscript deliver a compelling story. Detailed kinetic studies are combined with structural and computational studies. These studies together give good insight into enzyme functions.

I guess I would like to see direct measurements of affinity - ITC? This may enable the impact of mutation on the K_d of the complex to be assessed.

I also wondered whether more conservative mutations R to K may have been informative.

Aside from these concerns above - there are some minor things. The kinetics is complex and of course the K_m values here are apparent values. This is clear in the supplementary, but not in the main text. The HisZ concentration employed in the studies of ATPPRT may be critical, as the K_d values are different. I also note the analytical SEC indicates there may be other higher molecular weight species present.

Reviewer #3:

Remarks to the Author:

In this manuscript, Fisher et. al showed that mutations of HisGS with impaired reaction chemistry can be allosterically restored by HisZ despite that the HisZ:HisGS interface is located ~ 20 Å away from the active site. In the Arg56Ala-HisGS mutant, HisZ modulates the Arg32 dynamics so that it can partially compensate for the absence of Arg56. The study is of great interest but needs more details. In addition, there are numerous concerns to be addressed.

1. [Page 7, 1st paragraph] The authors claim that "MS/MS analysis of tryptic fragments proved the induction of D179N mutation". On the other hand, SI Fig. 2 shows the confirmation of all the mutants, what is the logic behind their claim, which only focuses on the D179N mutation?
2. [Page 7, 1st paragraph] The PRATP formation time for D179N & D179A mutations doesn't seem to be very different from the WT. Based on the presented data, the D179A mutation shows the potential for the PRATP formation, and it is recommended to mutate D179 into additional amino acids to check the potential of untested mutants.
3. [Page 8] Please clarify the statement "background noise of the assay".
4. [Page 8, Fig 1c and Fig 2a] The authors state that in the initial screening, these mutants change the catalytic activity, but the DSF-based thermal denaturation assays show similar behaviors. Detailed explanation is needed for these results. The results indicate that a clear difference between WT and the R32A mutant mainly exists in the absence of PRPP. In the presence of PRPP, the data still don't show similar behaviors. On the other hand, the mutants and WT showed clear differences. The authors need to explain the finding in more details.
5. [Page 8] The authors state that the "DSF-based thermal denaturation assays showed these mutants display similar thermal unfolding profiles to the WT protein". In the next paragraph, they further claim that the "the presence of PRPP increased the T_m of the mutants". Justification is needed.
6. [Page 10] The authors claim that "C115S led to a 117-fold reduction in activity (Table 1)", but the results showed no product detected (N.d*). Justification is needed.
7. [Page 11] The specific values for catalytic impairment should be provided to show the difference between the mutant and the WT, instead of fold difference in Table 1.
8. It is recommended to be consistent in the use of terms "activated state" and "inactivated state". Some paragraphs use "non-activated" but other paragraphs use "inactivated", which confuses readers.
9. [Page 16, last paragraph] The authors need to provide more details regarding the clear dynamical shift in the conformation, both graphically and statistically. Also, indicate in the correlation plot Fig 10 (SI) the targeted site of binding of PaHisZ of to PaHisGS. The binding of PaHisZ rescues the complex from both anti-correlated and correlated motion and bring them into static conformation. The authors also need to check the correlated motions for the mutant, both in the presence and absence of PaHisZ and compare the results with the WT.
10. [Page 16-17] The DCCM and the SPM results are conflicting with each other. The DCCM results show that binding of PaHisZ to PaHisGS brings the complex to the static position, but the SPM results show that the binding of PaHisZ increases intermonomer communication/interaction, which didn't support the DCCM results. More explanation is needed.
11. [Page 17] The authors claim that "comparing the SPM results, showed that PaHisZ binding increases the inter-monomer communication pathways across the two subunits of the PaHisGS dimer". The authors need to provide the pair-wise residue detail for SPM and the evidence for this pathway via mutation in any residues in both activated and non-activated states. Please label each residue in the pathway graphically and define the starting and ending point for the pathway. To validate the results, it is suggested that authors refer to Reference PMID: 31987943 for allosteric pathway analysis and also the reverse allosteric communication pathway.
12. [Page 17] The authors need to replace the dynamical flexibility indices (DFI) by residue fluctuation. The authors need to specify the targeted regions for clarity. Comparing both the correlation motion and SPM results, the DFI results seems quite different, where all residues fluctuate

highly. This doesn't agree with the results of correlation motion, where the author stated that the binding of protein brings the overall activated state into static position. Also, the residues no. of both DFI and DCCM is not the same. Some of the region fluctuates higher than the WT. Also, provide the RMSD results for these systems in different trajectories.

13. It is recommended to plot the free energy landscape by selecting the atomic distance (parameter-1) between residue Arg32/56 and the Pa of the PPi moiety of PRPP and the overall RMSD (parameter-2) of both activated and non-activated states, to check the conformational dynamics of both residues.

14. [Page 12] In comparison with the WT-PaATPPRT, KPRPP was unaltered by the mutations, and KATP was increased by a maximum of 4-fold. The k_{cat} was decreased by less than 2-fold for C115S-PaATPPRT as compared with WT-PaATPPRT, and by less than 4-fold and 6-fold for R32A- and R56A-PaATPPRT, respectively. Only R56A/K57A-PaATPPRT k_{cat} was reduced by more than one order of magnitude (~ 14 -fold) in comparison with WT-PaATPPRT, which is still a small effect in comparison with the 254-fold catalytic impairment of R56A/K57A-PaHisG. More justification is needed for the V_{max} and for the significance of mutations in this study.

15. [SI Fig. 1] The authors state that the SDS-PAGE of purified PaHisGS mutants eluted from the HisTrap FF column in the second chromatography. How to justify the separation of purified proteins after mutation?

Re: NCOMMS-22-21309

Reviewer #1

The authors note that while the WT holoenzyme shows little dependence on Mg vs Mn, the two Arg mutants show enhanced catalysis with Mn (Fig. 7). Their interpretation is that this shows that chemistry is rate contributing in the mutants but not WT. What they failed to identify, however, is the surprising outcome that the two mutants are substantially better catalysts than WT in the presence of Mn. In fact, R32A is 4–5-fold more active. The authors should make an attempt to explain the observation.

Author response: The reviewer is correct in their observation about the overall reaction rates.

Nonetheless, one must keep in mind that while the overall catalytic rates increase with Mn^{2+} for the mutants to levels higher than those for WT-*Pa*ATPPRT, these rates do not reflect the same reaction step. It is reasonable to assume that the rate of chemistry is drastically increased with Mn^{2+} for the WT-*Pa*ATPPRT, but that is not the dominant step when we measure steady-state rates. Instead, the rate we see is dominated by product release (which in our model is Mn-insensitive). With the mutants,

the increased rate we observe is the rate of chemistry itself. To make this important point clear to the readers, we have added the following under **Results/The R32A and R56A substitutions affect the chemical step**:

“Interestingly, R32A-*PaATPPRT* and R56A-*PaATPPRT* apparent first-order rate constants with Mn^{2+} are even higher than the corresponding one for WT-*PaATPPRT*. However, these rate constants do not reflect the same reaction step. It is likely that Mn^{2+} enhances the rate of chemistry substantially for WT-*PaATPPRT*, but the observed rate constant is dominated by product release.¹⁸ On the other hand, with R32A-*PaATPPRT* and R56A-*PaATPPRT*, the effect of Mn^{2+} is the enhancement of the rate of chemistry itself.”

On p. 20, it is stated, “these simulations reinforce the proposed role of R32 in stabilization of the leaving group at the transition state.” If I’ve understood the simulations properly, they were performed on the Michaelis complexes and therefore any stabilization identified in these experiments is relevant in the ground state. Perhaps these interactions become even more important during the chemical step, but such information does not come from these experiments. I think some careful rewording should clarify the authors’ intent.

Author response: The is a very good point. To avoid being unintentionally misleading in our statement, we modified sentence in question under **Results/Insights into the dynamics of allosteric rescue of R56A-*PaHisG_S***, which now reads only as:

“These simulations offer a dynamics-based hypothesis for the allosteric rescue of R56A-*PaHisG_S*: in the absence...”.

In the abstract, delete the hyphen between 20 and Å. Also, an en dash should be used instead of a colon for “protein–protein interactions”.

Author response: Done.

On p. 4, delete the hyphen between ribosyl and 1. Like other sugar phosphates, there are two separate words.

Author response: We deleted that hyphen, but then changed the phrase to “5-phospho- α -D-ribose 1-pyrophosphate”.

On p. 4, delete the comma between holoenzyme and allosterically. Commas should generally not separate the subject and predicate.

Author response: Done.

On p. 6, “structure” should be added after “PaATPPRT” at the end of the first sentence.

Author response: Done.

On p. 6, it states, “the multiple-turnover pre-steady-state rate-constant was in agreement with k_{cat} , suggesting chemistry is rate-limiting in the nonactivated enzyme reaction.” Add a comma between “multiple-turnover” and “pre-steady-state” (they are coordinate adjectives); remove the hyphen in “rate-constant”; and remove the hyphen in “rate-limiting” (a hyphen is only used when these two words function as a compound modifier immediately preceding a noun/noun phrase, as in “rate-limiting step”). Rate-limiting also appears at the end of the subsequent sentence in this paragraph.

Author response: Done.

On p. 7, “molecular mass” should be “molecular masses”, as there are multiple proteins, each with a unique mass. Correspondingly, at the end of the same sentence, change “predicted value” to “predicted values”.

Author response: Done.

On p. 7, the last sentence of the first paragraph is complex (it contains three main clauses). I recommend splitting into two, beginning the second with “PRATP formation was readily...”

Author response: Done.

On p. 7, second paragraph, it is stated, “The kcat for the mutants were reduced...” kcat is singular, so in order to agree with “were”, a plural term needs to be added, such as “kcat values”.

Author response: Done.

In Fig. 2c, the vertical scale is in units of $\mu\text{M s}^{-1}$, which is a rate. Therefore, it should be v , not [PRATP]. The same is true in Fig. 3a.

Author response: The y-axes in Fig. 2c and 3a are in fact rates, and we have replaced the [PRATP] for \$v\$ on both panels. The authors thank the reviewer for drawing our attention to this mistake.

In the legend to Fig. 2d, change “Close-up” to “Close-ups”.

Author response: Done.

On p. 13, last sentence, “thus” is an adverb, not a conjunction, so it requires a semicolon before it (not a comma), and a comma after it.

Author response: Done.

In Fig. 3c legend, provide the PDB code for the crystal structure, even though it may appear elsewhere in the document.

Author response: Done.

In several locations (listed below), the definite article is used when it ordinarily would not and should be deleted.

a. p. 15, penultimate line, “absent in the R56A-PaHisGS (Fig. 4).”

Author response: Done.

b. p. 17, last line, “the region containing the R32 and R56.”

Author response: Done.

c. Fig. 6 legend, “between C ζ of either R56 or R32 and the P α of the PPI moiety.” Here, “the” could be used but then it should appear before C ζ for consistency. Personally, I would not use “the” because P α is unambiguous. Whatever option is chosen should be made consistent with the text that appears near the end of the first sentence on p. 19.

Author response: We chose to remove the definite article before P \$\alpha\$ in the legend to Fig. 6 and before C \$\zeta\$ and P \$\alpha\$ near the end of the first sentence on now page 20.

In Fig. 4 legend, it should read, “The A56 side chain is contributed by one of the adjacent PaHisGS subunits. In the next line, Mg²⁺ is depicted as “a sphere” (add the indefinite article).

Author response: Done.

On p. 17, one of the sentences begins, “Interestingly, the calculated SPM (Fig. 6)”, but Fig. 6 does not contain any SPMs. Perhaps Fig. 5d was intended. In the next sentence, DFI is defined as “dynamical flexibility indexes” but in the supplementary information, “indices” is used instead. Choose one and use it consistently.

Author response: We replaced (Fig. 6) in the sentence in question for (Fig. 5d). We thank the reviewer for drawing our attention to the mistake. In the following sentence, we now use “dynamical flexibility indices” as in the Supplementary Information.

On p. 19, 3rd and 4th sentences read, “It can be gleaned from the data that the R56 side chain displays a bimodal distribution of distances in WT-PaHisGS (Fig. 6b). These comprise...” The plural pronoun “these” has not been properly defined. The authors are referring to what appears to be the two distances of maximal frequency in the bimodal distribution, but these positions have not yet been identified in the text. Otherwise, the only plural term in the preceding sentence is “distances”, but in that context, it is the entire domain, not just the two points, that is being described.

Author response: We agree with the reviewer. We changed the 4th sentence on now page 20 to “This distribution comprises...”, since it is the bimodal distribution referred to in the previous sentence that contains two peaks.

In Fig. 6a, the two arginines are rendered as free amino acids, but they should be residues within a peptide chain. Also, the Å symbol has no meaning here (the adjacent figures already provide the unit).

Author response: Fig. 6a was modified to reflect the proposed changes.

On p. 21, the apparent first-order rate constants with Mn²⁺ are compared with those for R32A and R56A holoenzymes, which are both larger. However, in parentheses, the values are described as being higher in one case and lower in the other.

Author response: The values in parentheses should both state “higher”. We have corrected this for the R56A-*Pa*ATPPRT case. We thank the reviewer for drawing our attention to the mistake.

On p. 25, in the first appearance of an organism’s name, the genus should be provided in full. Thus, it should be *Escherichia coli*.

Author response: Done.

The protease inhibitor cocktail has been written with a Scandinavian letter ø, but this is not correct. Either write it as Complete or cOplete, which is how Roche stylizes it.

Author response: We have now changed it to “Complete”.

On p. 26, first line, tricine is the last item in the list and therefore requires “and” before it and no comma after it.

Author response: Done.

On p. 30, it should be “after which 100 µL of D2O was added” (singular verb form). In the next sentence, change “spectrophotometer” to “spectrometer”.

Author response: Done.

On p. 32, first line, delete the comma after PRPP.

Author response: Done.

On p. 34, in eqs 3 and 4, the first set of brackets should be removed, as only addition is being performed. When comparing the radicands, the two sets of brackets are different sizes; perhaps typed parentheses were used in eq 3, while brackets were used in eq 4.

Author response: All fixed.

On p. 42, “Conceived” should be lowercase.

Author response: Done.

Reviewer #2

These authors previous works have proposed a mechanism for the activation of the catalytic domains by HisZ, organisation of charged functionality to facilitate loss of diphosphate leaving group - this is established. Is it therefore that surprising that significantly impaired mutants in the catalytic site, also are activated by HisZ? It feels a little as if the activations maybe somewhat overstated because of the very low rates of catalysis without HisZ present. Inclusion of the HisZ does improve the rates, but there is still impairment, relative to the wild-type enzyme. As is to be expected.

Author response: It could be expected that some of the catalytically impaired HisG mutants would be activated to a certain extent, just as the WT-HisG is, but this activation should be proportional to that of WT-HisG. For instance, R56A- or R32A-HisG are 42- and 25-fold less active than WT-HisG. This means that if binding of HisZ to WT-HisG activates catalysis by ~29-fold, then binding of HisZ to R56A- or R32A-HisG should be expected to activate catalysis by ~29-fold in each case, and the R56A-ATPPRT and R32A-ATPPRT mutants should be ~42- and ~25-fold less active than WT-ATPPRT. However, that is not what we observe. HisZ binding to the mutant HisGs disproportionately activates catalysis, 250-fold in the case of R32A-HisG and 250-fold in the case R56A-HisG, for example. There is no *a priori* reason to expect that. In fact, HisZ activation of the mutants is so astounding (in comparison with the ~29-fold activation of WT-HisG) that it in fact resurrects an otherwise severely compromised catalytic subunit to catalytic levels only 2- to 6-fold lower than the

WT-ATPPRT. This is even more surprising when one considers that HisZ does not bind anywhere

near the active site of HisG, ruling out any direct chemical rescue of the mutants. This activity rescue is purely allosteric.

Also while the allosteric rescue of the R32 and R56 mutants makes sense in terms of the proposed mechanism for the HisZ activation of the enzyme, the activation fold is most significant for the C115S mutant. This results does not seem to be commented on at length.

Author response: The reviewer is correct on the facts. Nonetheless, the difference between C115S and the other rescued mutants is that in the case of R32A and R56A mutants, “removal” of the guanidinium side chain causes the loss of activity in nonactivated mutants, which is rescued by HisZ, while in the case of C115, “removal” of the thiol side chain has only a minor effect on activity, as evidenced by the nonactivated C115A-HisG catalytic rate (Supplementary Table S2). This suggests a very modest importance for C115 in catalysis and does not support this residue’s role as a catalytic base (which we initially hypothesized based on our previous crystal structures). This latter point is already made in the manuscript under **Results/C115A-, D179A-, and D179N-*PaHisG*s are catalytically active.**

The severe catalytic impairment of C115S-HisG is therefore concluded not to be a result of the loss of the thiol, but the “introduction” of a hydroxyl group instead. We do not know the exact reason for the detrimental effect on catalysis caused by the serine residue at the 115 position, but we do hypothesize in the manuscript, page 10:

“Even though C115 is only modestly important for catalysis, its replacement by serine led to a 117-fold reduction in activity (**Table 1**), perhaps due to the introduction of a detrimental interaction.”

Our hypothesis is that allosteric rescue of the C115S-HisG by HisZ may be caused by HisZ binding disrupting the putative detrimental interaction introduced by S115. To bring this point across in the manuscript, we have now added under **Results/*PaHisZ* allosterically rescues C115S-, R32A-,**

R56A-, and R56A/K57A-*PaHisG*s catalysis:

“The k_{cat} decreased by less than 2-fold for C115S-*Pa*ATPPRT as compared with WT-*Pa*ATPPRT (it is possible *Pa*HisZ binding allosterically disrupts a putative catalytically detrimental interaction involving S115), ...”.

I guess I would like to see direct measurements of affinity - ITC? This may enable the impact of mutation on the K_D of the complex to be assessed.

Author response: We have long tried to measure the direct binding of HisZ to HisG (WT variants of both proteins) using ITC and fluorescence spectroscopy, but alas, we were never successful. With ITC, while there was an obvious qualitative heat exchange upon titration of one subunit into the cell containing other, the trace would not stabilize, and the ΔH vs molar ratio plots gave complex patterns that were not reproducible from one experiment to another. We deemed this to be possibly a consequence of a dynamic equilibrium between the hetero-octamer and other oligomeric states and/or transient disruption of one or more of the many interactions at the HisG/HisZ, HisZ/HisZ, and HisG/HisG interfaces. With fluorescence spectroscopy, we attempted to titrate HisG onto HisZ and follow the HisZ tryptophan fluorescence change. Here, experiments were plagued by a large and nonlinear inner filter effect of HisG even at relatively low concentrations.

Most importantly for this work, the indirect measurement of affinity of HisZ for WT and mutant HisGs can be assessed and compared across variants using the HisZ-induced increase in catalytic activity. We refer to the resulting K_D as “apparent K_D ” because measurements are made in the presence of both substrates (Figure 2c) and the readout is not directly binding, but activity increase. However, in all experiments where the affinity of the complex was important, for instance to assess *Pa*ATPPRT concentration to calculate k_{cat} , both substrates are in fact present, and activity increase is a direct consequence of HisZ binding, since the concentration of HisG we use in the experiment is too low for product formation to be detected in our spectrophotometric assay in the absence of HisZ. Furthermore, because all apparent K_D s were determined under the same conditions, the effect of mutation on the affinity of the complex is easily compared.

I also wondered whether more conservative mutations R to K may have been informative.

Author response: This is an excellent idea, and we thank the reviewer for suggesting it. We have carried out the experiment, focusing on R56 owing to its better understood role in catalysis. We have added the following to the manuscript.

Under **Results/The R32A and R56A substitutions affect the chemical step:**

“Given the proposed role of R56 in stabilising the departure of the negatively charged leaving group, the possibility that a lysine residue could replace R56 with similar catalytic ability was considered. To evaluate this possibility, R56K-*PaHisG_S* was produced. ESI/TOF-MS analysis resulted in the expected mass (**Supplementary Fig. 14a**), and DSF showed the mutation does not change the T_m of the protein (**Supplementary Fig. 14b**). At substrate concentrations saturating for WT-*PaHisG_S*, however, the R56K-*PaHisG_S* reaction rate is reduced ~24-fold in comparison with the WT-*PaHisG_S*, and although there is an ~11-fold activation in the presence of *PaHisZ*, the R56K-*PaATPPRT* reaction rate is still ~2-fold lower compared with the nonactivated WT variant (**Supplementary Fig. 14c**). These observations indicate the amino group cannot substitute for the guanidinium group at position 56 of *PaHisG_S*. Furthermore, allosteric activation by *PaHisZ* cannot rescue catalysis in this case.”

Under **Discussion**, at the end of the second paragraph:

“Interestingly, replacement of a key arginine residue for a lysine was detrimental to catalysis in adenylate kinase as well.⁴⁶”

Under **Methods/Protein expression and purification:**

“..., R56K, ...”

Under **Methods/Activity of *PaHisG_S* mutants in the absence of *PaHisZ*:**

“Exceptionally, activity of R56K-*PaHisG* (6 μ M) was assayed both in the absence and presence of *PaHisZ* (20 μ M).”

Supplementary Fig. 14 was added to the Supplementary Information. Subsequent supplementary figures were renumbered accordingly in the main manuscript and in the Supplementary Information file.

The sequence of the primers for generating the R56K mutation were added to now **Supplementary Table 9**.

The kinetics is complex and of course the K_m values here are apparent values. This is clear in the supplementary, but not in the main text.

Author response: We agree with the reviewer that all steady-state kinetic parameters we measured in this work are apparent values, because one substrate was varied at a fixed, albeit saturating, level of the co-substrate, not against an array of concentrations of the co-substrate.

That is why we stated such from the outset, on page 7:

“...WT-, C115A-, D179A-, and D179N-*PaHisG_S* obeyed Michaelis-Menten kinetics (**Fig. 1d**), and data fit to equation (1) produced the apparent steady-state kinetic parameters...”

On page 12:

“WT-, C115S-, R32A-, R56A-, and R56A/K57A-*PaATPPRT* obeyed Michaelis-Menten kinetics (**Fig. 2e**), and data fit to equation (1) produced the apparent steady-state kinetic parameters...”

On page 15:

“WT-*PaHisG_S/AbHisZ* and R56A-*PaHisG_S/AbHisZ* hybrid complexes obeyed Michaelis-Menten kinetics (**Fig. 3b**), and data fit to equation (1) produced the following apparent k_{cat} , K_{PRPP} , and K_{ATP} ...”

To reflect this point also in Table 2, the title to this table now reads:

“**Table 2.** Apparent steady-state kinetic parameters (mean \pm fitting error) for *PaATPPRT* variants (all mutations are in *PaHisG_S*).”

We also added “apparent” to the text on page 13, which now reads:

“...apparent steady-state kinetic parameters (**Supplementary Fig. 6**)...”

The HisZ concentration employed in the studies of ATPPRT may be critical, as the K_d values are different.

Author response: We agree with the reviewer. We use equation 4 to calculate the concentration of the multi-protein complex based on each apparent K_D and the concentrations of the two individual proteins involved. Moreover, the *PaHisZ* or *AbHisZ* concentrations used for kinetic analysis of the WT and mutant variants are in the respective plateau region as indicated in Fig. 2c, Fig. 3a, and Supplementary Fig. 6a. This means HisZ is saturating and therefore there should be only negligible amounts of free *PaHisG* in the experiments where *PaATPPRT* is expected to be the enzyme form. Furthermore, we know from this and our previous work on this system that such low amounts of *PaHisG* produces no detectable rates above our assay background noise.

I also note the analytical SEC indicates there may be other higher molecular weight species present.

We agree with the reviewer, and the point had already been made in the legend to Supplementary Fig.

3. We would also like to note that similar SEC profiles are obtained for the impaired mutants.

Nonetheless, to make the reviewer's point more explicit in the main manuscript, we changed the penultimate sentence on page 8 to:

“Analytical size-exclusion chromatography produced similar elution profiles for WT-, C115S-, R32A-, R56A-, and R56A/K57A-*PaHisG*s (Supplementary Fig. 3), which includes the expected dimer²⁹ and a higher oligomeric state.”

Reviewer #3

[Page 7, 1st paragraph] The authors claim that “MS/MS analysis of tryptic fragments proved the induction of D179N mutation”. On the other hand, SI Fig. 2 shows the confirmation of all the mutants, what is the logic behind their claim, which only focuses on the D179N mutation?

Author response: The introduction of desired mutation(s) had already confirmed at the DNA level by DNA sequencing of all mutants. We further confirmed the mutations for all variants at the protein level by MS. The difference is that for almost all mutants, the intact mass was detected, which

matched the expected mass of each mutant. However, for the D179N, we instead used tryptic

digestion followed and peptide identification by MS/MS. This was stated clearly on page 7:

“Electrospray ionisation/time-of-flight-mass spectrometry (ESI/TOF-MS) confirmed the molecular masses of WT-, C115A-, C115S-, D179A-, R32A-, R56A-, and R56A/K57A-*PaHisG_S* variants were in agreement with the predicted values (**Supplementary Fig. 2**). The introduction of the D179N mutation was confirmed by MS/MS analysis of tryptic fragments (**Supplementary Fig. 2**).”

The logic behind this is that N sometimes may undergo hydrolysis to D in proteins. Though this is not common, we wanted to use a more direct method (identification of the specific tryptic peptide carrying the mutation) to confirm mutation in this case.

[Page 7, 1st paragraph] The PRATP formation time for D179N & D179A mutations doesn't seem to be very different from the WT. Based on the presented data, the D179A mutation shows the potential for the PRATP formation, and it is recommended to mutate D179 into additional amino acids to check the potential of untested mutants.

Author response: We agree with the reviewer that D179N and D179A are not very different from the WT in terms of PRATP formation ability, that was our conclusion as well, as stated in the manuscript. They do not show only potential for PRATP formation, these variants can definitely catalyse formation of PRATP with only small reduction in catalytic ability in comparison with the WT. This is why we concluded that D179's role in catalysis was only modest. As discussed in the manuscript, we hypothesized that D179 could be the catalytic base, a hypothesis we subsequently ruled out since neither the D179A nor the D179N variants can mediate acid-base catalysis but both are still robustly active. Based on these data, mutation of D179 to cover the remaining 17 amino acid alternatives is not just time-consuming and labour-intensive, but would not add to the conclusions of this manuscript considering the aforementioned very modest importance of D179 to catalysis and substrate binding.

[Page 8] Please clarify the statement “background noise of the assay”.

Author response: Any quantitative experimental assay, particularly those based on spectroscopy as is

the case here, have a limit of detection. This limit is the minimum amount PRATP (to focus only on our case, which is the relevant one for this discussion) that can be reliably detected by the spectrophotometer above the “noise”, *i.e.* the signal oscillation of the negative control (in the absence of enzyme in our case). How low this limit is will usually depend on the sensitivity of the spectrophotometer and on the extinction coefficient of the product being detected at a given wavelength. Together, these parameters form the sensitivity of the assay. This signal oscillation is routinely referred to in the experimental community as background noise. It would be incorrect to state, in our case, that a given enzyme variant did not generate product. All we can say is that it did not generate enough product to be detected given above the background noise of our assay. To avoid any doubts, we have modified the text in question on page 8 to:

“...background noise of the assay (no-enzyme control)...”

[Page 8, Fig 1c and Fig 2a] The authors state that in the initial screening, these mutants change the catalytic activity, but the DSF-based thermal denaturation assays show similar behaviors. Detailed explanation is needed for these results. The results indicate that a clear difference between WT and the R32A mutant mainly exists in the absence of PRPP. In the presence of PRPP, the data still don't show similar behaviors. On the other hand, the mutants and WT showed clear differences. The authors need to explain the finding in more details.

Author response: The mutants referred to on page 8 do display reduced catalytic activity. This is demonstrated experimentally in Fig. 1c and 2b, which measure product formation in our enzyme activity assay. Fig. 2a shows DSF-based thermal denaturation profiles for WT and mutant variants. This assay does not measure catalytic activity. Instead it is used here (top panel) to show that the mutants follow a similar denaturation profile to the WT enzyme, meaning the mutations are not reducing activity because they destabilise the tertiary structure of the protein. Yes, the R32A mutation even causes an ~3-°C thermal stabilisation of the protein in relation to the WT version. To make this conclusion clearer in the manuscript, we have modified the sentence on page 8 to the following:

“DSF-based thermal denaturation assays showed these mutants display similar thermal unfolding profiles to the WT protein (Fig. 2a), demonstrating the mutations do not thermally destabilise the tertiary structure of the protein, ...”

We have previously reported that *PaHisG_S* follows a strictly ordered kinetic mechanism where PRPP binds to the free enzyme, followed by ATP binding, and we have also shown that PRPP binding causes an increase in thermal stabilisation of the WT protein as evidenced by an increase in the T_m increase in DSF assays (see references 18 and 29 of the manuscript). Hence we used this as a qualitative indicator to show (bottom panel) that not only do the mutants demonstrate similar thermal stability to WT, but also they can bind PRPP as expected based on the kinetic mechanism of the enzyme, as evidenced by the increase of the T_m for all catalytically compromised variants in the presence of PRPP (Supplementary Table 3). This latter point is already clear from the text on page 8.

[Page 8] The authors state that the “DSF-based thermal denaturation assays showed these mutants display similar thermal unfolding profiles to the WT protein”. In the next paragraph, they further claim that the “the presence of PRPP increased the T_m of the mutants”. Justification is needed.

Author response: For the overall significance and conclusions from the DSF assay, please refer to the answer to the previous comment. What we stated on page 8 is that the presence of PRPP increases the T_m of the mutants as it does the WT's, as expected given our previous results. The justification for the specific assertion that PRPP binding increases the T_m of the mutants and WT is in our data (Fig. 2a, Supplementary Table 3), and the explanation is in the manuscript text and answer to the comment above.

[Page 10] The authors claim that “C115S led to a 117-fold reduction in activity (Table 1)”, but the results showed no product detected (N.d*). Justification is needed.

Author response: Product is detected for the C115S-*PaHisG* in our extended-time PRATP detection assay as shown in Fig. 2b, where there is clear product formation for this mutant above the no-enzyme control. This point is clearly stated in the last sentence on page 8. The 117-fold reduction in rate

comes from the linear regression of the product formation data, yielding the apparent rate constant shown in Table 1.

The specific values for catalytic impairment should be provided to show the difference between the mutant and the WT, instead of fold difference in Table 1.

Author response: Catalytic impairment can only be evaluated in relation to a standard, in this case the standard is the apparent first-order rate constant (v/E_T) for the WT enzyme. We do report the v/E_T values for each mutant in Table 1, along with the corresponding values for catalytic impairment. As explained in the footnote in Table 1, catalytic impairment (-fold change) is not a difference, but a ratio of v/E_T for the WT enzyme (given in Supplementary Table 1) to v/E_T for each mutant. It seems the only other parameter we may show explicitly in Table 1 is the v/E_T for the WT, which we have now shown in the footnote instead of just referring the reader to Supplementary Table 1.

It is recommended to be consistent in the use of terms “activated state” and “inactivated state”. Some paragraphs use “non-activated” but other paragraphs use “inactivated”, which confuses readers.

Author response: We have replaced “inactivated” by “non-activated” at the bottom of page 16, the only instance where inactivated was used in the manuscript. We thank the reviewer for directing our attention to that.

[Page 16, last paragraph] The authors need to provide more details regarding the clear dynamical shift in the conformation, both graphically and statistically. Also, indicate in the correlation plot Fig 10 (SI) the targeted site of binding of PaHisZ of to PaHisGS. The binding of PaHisZ rescues the complex from both anti-correlated and correlated motion and bring them into static conformation. The authors also need to check the correlated motions for the mutant, both in the presence and absence of PaHisZ and compare the results with the WT.

Author response: We thank the Reviewer for bringing our focus back to these plots. We were also confused by the original results, as one would expect the allosteric activation to increase not reduce ordered motions in the system, but we checked the DCCM calculation many times and everything

appeared to be correct. We had commented on the fact that this is surprising in the original submission at the bottom of page 16 of the original version: “However, it is surprising to observe a loss in correlated motions upon allosteric activation, which suggests that binding of *PaHisZ* restricts *PaHisG_S* motions.”

Upon further examination of the literature, it seems that how the fitting is done (in this case whether to the dimer or the full hetero-octamer) can have significant impact on the results, see discussion in [Hünenberger et al., J. Mol. Biol. 252 (1995), 492]. This does not impact our analysis of *PaHisG_S*, where we had anyhow performed the fitting to the full dimer, but we realised from reading this paper that the fitting could potentially impact our DCCM analysis after binding of *PaHisZ*. Because of this, and taking into account the paper mentioned above, we have revisited our analysis and observed that although the correlation matrix is extracted only on the dimer of the hetero-octamer, whether the fitting of the trajectory (to remove rigid-body motions) was done on the dimer (subset of atoms) or the whole hetero-octamer, impacts our results. In the original submission, we had fitted our trajectory to just the dimer, and based on careful reading of the Hünenberger paper we realized that this choice was hiding relevant fluctuations, which are actually critical to understanding the allosteric activation through *PaHisZ* binding. Thus, we have refitted the trajectory to the full hetero-octamer and performed the fluctuation analysis on the dimer and updated the manuscript accordingly.

Specifically, we have updated Supplementary Fig 10 with the new data for both the wild-type and R56A variants, in their non-activated and activated forms. As requested by the reviewer, we have also now highlighted the increased intermonomer correlations, as well as the important regions in the communication pathway, on the DCCM plots. Based on the new data, it can be seen that binding of *PaHisZ* increases both intra- and intermonomer ordered motions in the system, in line with the SPM

results. Furthermore, as requested by the reviewer, we have included also analysis of the R56A variant, and it can be seen that here the global effects are similar to the wild-type.

We have also calculated the Spearman's rank correlation coefficients to statistically measure the similarity between the correlation matrices for the non-activated and activated systems, as requested by the Reviewer, based on the corrected DCCM matrices with fitting to the full hetero-octamer. The ρ values between the non-activated and activated systems in the wild-type and R56A variants are 0.26 and 0.37, suggesting low similarity between these two pairs of systems, whereas the analogous comparison between non-activated and activated wild-type/R56A pairs is 0.86 and 0.84, respectively. We have updated the associated discussion accordingly on pages 16-17 and updated the raw data of the correlation matrices in the ZENODO package associated with the work. The updated text now reads:

Page 16: "...for both WT and R56A variants."

Page 17: "...increases in..."

Page 17: "and similar effects upon activation of both the wild-type enzyme and the R56A variant.

Spearman's rank correlation coefficients between the non-activated and activated systems are 0.26 for WT-*PaHisG_S* and 0.37 for R56A-*PaHisG_S*. In contrast, the coefficients between WT-*PaHisG_S* and R56A-*PaHisG_S* in each of the non-activated and activated states are 0.86 and 0.84, respectively. This suggests high similarity between WT-*PaHisG_S* and R56A-*PaHisG_S* when the two systems are in the same state (non-activated vs. activated), but low similarity between the non-activated and activated states of each individual variant. It is expected that external structural perturbations (such as ligand binding or, in this case, the binding of *PaHisZ* to *PaHisG_S*) would alter such conformational fluctuations, as has been observed in other allosteric systems,³⁶⁻³⁸ and it can also be seen here that *PaHisZ* binding increases order in the system."

Page 35: “Two-sided Spearman’s rank order correlation tests were used to measure the monotonicity of the relationship between distinct correlation matrices. The upper triangle of each 416×416 matrix was used, representing a sample size of $n = 86,528$. The standard assumptions (data measured on an interval scale, and the two variables are monotonically related) were used.”

[Page 16-17] The DCCM and the SPM results are conflicting with each other. The DCCM results show that binding of PaHisZ to PaHisGS brings the complex to the static position, but the SPM results show that the binding of PaHisZ increases intermonomer communication/interaction, which didn’t support the DCCM results. More explanation is needed.

Author response: Please see the response to point 9, and updated discussion. Thanks to the Reviewer’s comments we have repeated the DCCM analysis with a better fitting procedure, and the new DCCM results are consistent with the SPM results.

The authors claim that “comparing the SPM results, showed that PaHisZ binding increases the intermonomer communication pathways across the two subunits of the PaHisGS dimer”. The authors need to provide the pair-wise residue detail for SPM and the evidence for this pathway via mutation in any residues in both activated and non-activated states. Please label each residue in the pathway graphically and define the starting and ending point for the pathway. To validate the results, it is suggested that authors refer to Reference PMID: 31987943 for allosteric pathway analysis and also the reverse allosteric communication pathway.

Author response: We want to thank the Reviewer for this valuable suggestion. We have now included the pair-wise residue detail for the SPM in **Supplementary Tables 5 – 7**. We want to also note that while building the tables we discovered that the 3D representation of the SPM for the non-activated and activated dimer was shifted by three positions at the second monomer due to the presence of the ligands in the PDB structure that were not considered for the $C\alpha$ -matrix analysis. We have now updated Figure 5 accordingly, as well as updating Supplementary Figure 10 based on the updated

correlation matrix with the improved fitting procedure thanks to the Reviewer's previous comment.

We would like to highlight that such changes do not modify the conclusions drawn where the activated dimer displays a tighter network and the nonactivated shows a "broken" network of interactions. We have now also labelled V257, G268, and I269 from the histidine binding loop on Figure 5, as well as a few other residues from the node-weakening analysis we performed. We do appreciate the Reviewer's suggestion and we agree in that, information on the residues of the shortest path map is beneficial for the reader and so we have included the pair-wise residue detail where the map of residues can be easily tracked, but we did not further label more residues on Figure 5 for clarity, to avoid crowdedness on the figure.

Regarding the starting and ending point of the pathway, we would like to bring to the Reviewer's attention the detail that although the SPM (DOI: 10.1021/acscatal.7b02954) method we use in this work has many similarities with other methods also based on the shortest path approach, here we do not define or aim to find paths only from a pre-defined start point to an end point. Rather, Osuna's implementation in DynaComm.py computes the shortest path lengths with the Dijkstra algorithm, going through all nodes of the graph to identify the shortest path that goes from the first until the last protein residue. It therefore identifies which edges of the graph are shorter (that is, more correlated) and more frequently used for going through all residues of the protein. All edges are then normalized, and only those edges having the largest contribution are represented in the SPM (see DOI: 10.1002/wcms.1502 and 10.1021/acs.chemrev.5b00590). This has now been clarified in the discussion on page 17. Thus, while there is overlap between this approach and the strategy used in the paper pointed out by the reviewer, there are also significant methodological differences in how the paths are constructed, particularly as PMID 31987943 uses pre-defined start point for calculating the pathways, which requires a different type of validation for rigor.

We have modified the following in the manuscript.

Page 17: “...(SPM) approach as implemented by Osuna and coworkers,³⁹...”

Page 17: “As described by Osuna⁴¹ and Guo and Zhou⁴² in the implementation we have used, the shortest path lengths are computed using the Dijkstra algorithm by going through all nodes of the graph to identify the shortest path from the first to the last protein residue. This allows the identification of which edges of the graph are shorter (*i.e.* more correlated, see **Supplementary Tables 5 – 7**) and more frequently used for going through all residues of the protein. All edges are then normalized, and only those edges having the largest contributions are represented in the SPM.”

Page 17: “...computed...”

Following from this, we have nevertheless performed node weakening analysis, following the Reviewer’s suggested Reference PMID: 31987943, related reference DOI: 10.1038/srep31005 and also DOI: 10.1073/pnas.0810961106. Specifically, we have removed the nodes corresponding to the residues located at the contact area between HisG_S and HisZ where the SPM goes through the two proteins (5 nodes on HisG_S and 7 nodes on HisZ), and calculated the Δ CPL (characteristic path length) upon removal of each node, since change in CPL upon removal of a node is a measure of its effect on the communication within the network. Based on this, we have selected three residues (Y105 in HisG_S and N185 and K186 in HisZ), which display the largest Δ CPL and performed molecular dynamic simulations of four selected mutants (Y105A, Y105F, N185A and K186D). However, as pointed out by Luthey-Schulten and co-workers, (DOI: 10.1073/pnas.0810961106) the results might be difficult to interpret due to its sensitivity to the geometry of the network. Our results show that the SPM is slightly reorganized upon removal of a node, without displaying critical changes, either in the static picture upon removal of the corresponding nodes, or on the SPM analysis on the mutated systems. This is now discussed on page 18 and Supplementary Figures 12 and 13. These results are in line with our assumption that the allosteric activation is due to a global nesting effect of HisZ over HisG_S, with a preferred but not unique allosteric activation pathway.

We have updated the Methodology, Results and SI of the manuscript to incorporate the corresponding analysis as follows:

Page 18: “To identify residues that have the largest effect on allosteric signal communication, we have performed node-weakening analysis by removing the nodes corresponding to the residues located at the interface between *PaHisG_S* and *PaHisZ*, where the SPM goes through both proteins (5 nodes on *PaHisG_S* and 7 nodes on *PaHisZ*), and calculating the change in CPL (characteristic path length) upon removal of each node (**Supplementary Table 8**).⁴⁴⁻⁴⁶ We then selected the three residues displaying the highest impact on the Δ CPL and performed molecular dynamic simulations of four selected *in-silico* mutants of *PaATPPRT* (Y105A-*PaHisG_S*, Y105F-*PaHisG_S*, N185A-*PaHisZ*, and K186D-*PaHisZ*), with the aim of disrupting the communication pathway between *PaHisG_S* and *PaHisZ*. When comparing the SPM of the WT- *PaATPPRT* with the various mutants we generated *in-silico*, we see that the pathway is slightly reorganized without displaying critical changes that disrupt the communication signal between the two proteins (**Supplementary Fig. 12**). These results are in line with a proposal that allosteric activation is due to a global “nesting” effect of *PaHisZ* over *PaHisG_S*, with a preferred but not unique allosteric activation pathway.”

Page 34: “Previously prepared WT-*PaATPPRT*:PRPP:ATP was used as starting point to introduce, *in silico*, the point mutations from the node-weakening analysis: Y105A-*PaHisG_S*, Y105F-*PaHisG_S*, N185A-*PaHisZ*, and K186D-*PaHisZ*. Corresponding rotamers were selected using the Dunbrack 2010 Rotamer Library.⁵⁹”

Last sentence of the legend to Fig. 5d: “The positions of key residues in the histidine-binding loop and some included in the node-weakening analysis are denoted by arrows; if they are adjacent to the SPM, the number in brackets indicates how many amino-acid residues away they are from the closest residue encompassed by the SPM.”

The authors need to replace the dynamical flexibility indices (DFI) by residue fluctuation. The authors need to specify the targeted regions for clarity. Comparing both the correlation motion and SPM results, the DFI results seems quite different, where all residues fluctuate highly. This doesn't agree with the results of correlation motion, where the author stated that the binding of protein brings the overall activated state into static position. Also, the residues no. of both DFI and DCCM is not the same. Some of the region fluctuates higher than the WT. Also, provide the RMSD results for these systems in different trajectories.

Author response: These different analyses are meant to provide different types of information. The DCCM plots provide insight into the nature of any (anti)correlated motions across the whole scaffold, the SPM analysis calculates allosteric communication pathways based on residue correlations, RMSF provides information about the flexibility of backbone $C\alpha$ -atoms during the simulations (in this case not so informative, see below) and, as described by Ozkan in her original paper (DOI: 10.1111/eva.12052), DFI measures the dynamic response at each residue in a protein when the system is perturbed (*e.g.* by a random Brownian kick). The latter is intended to mimic the natural condition in a crowded cellular environment, where the protein is exposed to many different random forces, and therefore, provided that the hypothesis that there is an underlying functionally important dynamical (fluctuation) profile for the 3D-structure of a protein is correct, the DFI analysis captures the contribution of each position in the protein to functional dynamics, highlighting key residues that mediate functionally important dynamical fluctuations. This was validated extensively in the aforementioned paper, and has been applied successfully in subsequent work, including in a recent protein design effort by Ozkan (DOI: 10.1038/s41467-021-022089-0). Thus, this provides important information to the paper. The RMSF analysis is far less informative and merely shows that there are slightly lower fluctuations in the activated system in the region in contact with *PaHisZ* ($\alpha 1$, $\alpha 8$ and loop containing R56) and greater fluctuations in the loop, not included in the allosteric path, containing residues ~150-155, for the activated system due to some fluctuating contacts with another

PaHisZ loop. We have now provided more background about the DFI analysis on page 18; however, we have not included the RMSF plot (below, included in this response letter for review purposes only) as it does not provide significant additional insight. Please note also that since we corrected the DCCM analysis, the different analyses are far more consistent.

We added the following on page 18: “...which measure the dynamic response at each residue in a protein when the system is perturbed (e.g. by a random Brownian kick). This mimics the natural condition in a crowded cellular environment, where the protein is exposed to many different random forces. Thus, the DFI analysis aims to capture the contribution of each position in the protein to the underlying functional dynamics, highlighting the key residues and regions that mediate functionally important dynamical information. Our analysis...”

Supplementary Fig 15 now show RMSDs for all trajectories as requested.

It is recommended to plot the free energy landscape by selecting the atomic distance (parameter-1) between residue Arg32/56 and the P α of the PPi moiety of PRPP and the overall RMSD (parameter-2) of both activated and non-activated states, to check the conformational dynamics of both residues.

Author response: We plotted the free energy landscapes suggested by the reviewer as a function of distance vs. RMSD, as suggested by the reviewer. The resulting plots are shown below for the non-

activated system; as can be seen from this data, plotting the distance vs. overall RMSD is

unfortunately not very informative compared to the distance analysis already provided in the paper.

[Page 12] In comparison with the WT-PaATPPRT, KPRPP was unaltered by the mutations, and k_{ATP} was increased by a maximum of 4-fold. The k_{cat} was decreased by less than 2-fold for C115S-PaATPPRT as compared with WT-PaATPPRT, and by less than 4-fold and 6-fold for R32A- and R56A-PaATPPRT, respectively. Only R56A/K57A-PaATPPRT k_{cat} was reduced by more than one order of magnitude (~14-fold) in comparison with WT-PaATPPRT, which is still a small effect in comparison with the 254-fold catalytic impairment of R56A/K57A-PaHisG. More justification is needed for the V_{max} and for the significance of mutations in this study.

Author response: We are not 100% sure of what the reviewer means here, and believe there has been a misunderstanding. Specifically, we can categorically state that statements quoted by the reviewer are fully justified by the data depicted in Fig. 2e, whose values are reported in Table 2. We would like to point out that $k_{cat} = V_{max}/E_T$; we plot v/E_T values instead of rates ($[product]/time$), such as in Fig. 2e, and therefore fitting to the Michaelis-Menten equation gives k_{cat} directly, not V_{max} . Obviously, whatever the effect of the mutations on k_{cat} , the exact same effect is exerted on V_{max} , since one is linearly dependent on the other.

[SI Fig. 1] The authors state that the SDS-PAGE of purified PaHisGS mutants eluted from the HisTrap FF column in the second chromatography. How to justify the separation of purified proteins after mutation?

Author response: The *PaHisGS* mutations did not influence the purification protocol whatsoever, and each variant was expressed and purified independently of the others. The SDS-PAGEs are just showing the raw data for the elution from the second round of affinity chromatography (as described in the figure legend) in which the desired protein (mutant or WT, is does not matter) leaves the nickel column in the flow through because the his-tag has been cleaved off by TEVP. This is a well established protocol which we used for WT and all the mutants, as indicated in the Methods section citing reference 28.

Reviewers' Comments:

Reviewer #1:

Remarks to the Author:

I am satisfied by the authors' changes. Publish as is.

Reviewer #2:

Remarks to the Author:

The manuscript has been revised according to the reviewers' comments. The text has been clarified and some new experimental detail has been included. This revision largely satisfies key concerns.

There is nice work here detailing the power of remote interactions on activity of the catalytic site. Indeed it does show that these interactions can compensate for mutations that result in catalytic impairment.

Reviewer #3:

None